# Reconciling the accuracy-diversity trade-off in recommendations

## ABSTRACT

When making recommendations, there is an apparent trade-off between the goals of *accuracy* (to recommend items a user is most likely to want) and *diversity* (to recommend items representing a range of categories). As such, real-world recommender systems often explicitly incorporate diversity into recommendations, at the cost of accuracy.

We study the accuracy-diversity trade-off by bringing in a third concept: user utility. We argue that accuracy is misaligned with user utility because it fails to incorporate a user's consumption constraints; at any given time, users can typically only use at most a few recommended items (e.g., dine at one restaurant, or watch a couple of movies). In a theoretical model, we show that utility-maximizing recommendations—when accounting for consumption constraints—are naturally diverse due to diminishing returns of recommending similar items. Therefore, while increasing diversity may come at the cost of accuracy, it can also help align accuracy-based recommendations towards the more fundamental objective of user utility. Our theoretical results yield practical guidance into how recommendations should incorporate diversity to serve user ends.

## CCS CONCEPTS

• **Information systems** → **Information retrieval diversity**; **Recommender systems**.

## KEYWORDS

Accuracy-diversity trade-off, recommender systems, theoretical modeling

**ACM Reference Format:**
Anonymous Author(s). 2018. Reconciling the accuracy-diversity trade-off in recommendations. In *Proceedings of Make sure to enter the correct conference title from your rights confirmation emai (Conference acronym 'XX)*. ACM, New York, NY, USA, 26 pages. https://doi.org/XXXXXXX.XXXXXXX

## 1 INTRODUCTION

Recommender systems are often built to maximize *accuracy*, the percentage of recommended items that a user likes. This objective is well-suited for the machine-learning-based algorithms in common use. Recommender systems also seek to heuristically incorporate *diversity*, since users empirically prefer to be shown items from a range of categories [6, 34, 39, 48]. In practice, however, these goals are in tension. To counter accuracy-maximization's tendency

toward homogeneity, real-world recommender systems inject diversity into recommendations using a range of heuristics [21, 37]. A wide literature proposes methods to address the apparent "accuracy-diversity trade-off" (e.g., [1, 2, 4, 5, 7, 12, 16, 17, 20, 22, 23, 25, 26, 28, 31, 33, 35, 36, 41, 43, 49, 52, 54, 55]).

Despite the practical importance of navigating the trade-off, a deeper understanding of the underlying tension is missing. Without a principled understanding of the trade-off, attempts to diversify recommendations have difficulty moving beyond intuition—and difficulty articulating what they are accomplishing at a deeper level.

*Reconciling the trade-off.* In this paper, we conceptualize and analyze the relationship between accuracy and diversity by observing that there are in fact three fundamentally distinct quantities of interest in these problems: accuracy, diversity, and user utility. None of these quantities serves as a proxy for any other, and a true understanding of the trade-offs requires understanding how all three of them interact. Moreover, the heart of the problem is really about maximizing one of these three—namely user utility, since this is what users actually experience. In particular, we argue that accuracy, in general, misrepresents user utility, and that by considering a better-conceived measure of utility, the trade-off with diversity dissipates. In turn, our results inform how diversity supports utility-maximizing recommendations.

Accuracy does correspond directly to a model of user utility in a very specific situation: when users obtain value from *all* recommended items—that is, with binary utility, value 1 for each item liked and 0 for each disliked. We argue that under more reasonable and general assumptions, however, accuracy is misaligned with user utility because it does not consider *consumption constraints*— limits on the number of recommended items a user can use. At a given time, a user can only watch one movie, dine at one restaurant, or purchase one new TV; a job recruiter can only extend interviews to a handful of recommended candidates.

A more precise measure of user utility accounts for consumption constraints and therefore focuses on the value the user obtains from the *best* items they are recommended. Given a "unit consumption constraint" and binary value for each item, this reduces to the probability that the user is shown *at least one* item they like. Once accounting for consumption constraints, we show that user utility is in fact aligned with and supported by diversity; a preference for diversity arises endogenously in our model. As a consequence, efforts to navigate the accuracy-diversity trade-off can be viewed not as balancing two genuinely competing desiderata, but rather as using diversity to steer accuracy-maximizing recommendations towards utility-maximizing recommendations.

To see why consumption constraints induce diversity, consider a thought experiment by Steck [50] in the design of recommendations. A user on a movie-streaming service is in the mood for comedy 80 percent of the time, and action 20 percent of the time. How many movies of each genre should we recommend? The items the user will like with the highest probability are mostly comedy, meaning that the accuracy-maximizing set of recommendations is

likely to be fairly homogeneous. But now suppose that we aim to maximize the probability the user likes at least one recommended movie—accounting for a "unit consumption constraint." Now, recommending only comedy movies is suboptimal. If the user is in the mood for action, they will be left without any options; meanwhile, it is not beneficial to recommend many additional comedy movies, since the user only cares that they have a single good movie to watch. In this way, accounting for consumption constraints intuitively induces diverse recommendations.

*A model of recommendations.* To make this intuition precise—that user utility is aligned with diversity—and to understand when and to what extent it holds, we need to analyze the diversity of accuracy- and utility-maximizing recommendations. (From here on, utility refers to a measure that accounts for capacity constraints.) A primary contribution of our work is a stylized-but-rich model of recommendations that is analytically tractable in this respect.

In our model, items of varying quality belong to discrete types, and each user has a probability distribution over types. In a given session, the user is in the mood for one of these types, where the type is drawn from the distribution. (Uncertainty of a user's mood can arise either due to genuine stochasticity, or limitations in the recommender's inferential abilities.) This model lends itself to an interpretable measure of diversity, where a set of recommendations is diverse if it represents items from many types roughly equally.

We derive results in an asymptotic regime where the number of recommendations grows large, obtaining precise characterizations of accuracy- and utility-maximizing recommendations as a function of model parameters that control the quality of items within and across types. We show in computational experiments that our theoretical findings hold more generally.

*Diminishing returns drive our results and proof technique.* Our results connect the composition of recommendation sets with the rate of diminishing returns when recommending more items of a given type (with respect to accuracy or utility). With large diminishing returns in one type, after recommending a few items of that type, a recommender becomes incentivized to recommend from other types. Roughly speaking, utility induces sharper diminishing returns than accuracy, and thus more diversity. The key steps in our proofs are to (1) precisely characterize the asymptotic behavior of these diminishing returns under different parameterizations of our model, and (2) to show how this behavior determines the asymptotic representation of item types in optimal recommendations.

*Overview of results.* In a basic setting (Theorem 1), the model confirms our intuition in a striking way. In this setting, accuracy-maximizing recommendations are completely homogeneous (representing only items from one type); yet, by accounting for consumption constraints, utility-maximizing recommendations are completely diverse (representing each type with equal proportion) in the limit. This uncovers a surprising fact—that even if the user prefers one type with higher probability than another, the optimal set of recommendations may contain an equal proportion of each.

In a more general setting (Theorems 2a and 2b), we consider differences in item quality within and across types, accounting for the idiosyncratic properties of recommendation settings.

Theorem 2a shows that accuracy-maximizing recommendations become more diverse when item quality decays at a faster rate (i.e., the recommender quickly begins to run out of "good options"). This accords with our conceptual understanding that larger diminishing returns induces more diversity. However, when this rate of decay is "reasonable" (in a sense made precise in Section 3.2), accuracy-maximizing recommendations remain relatively homogeneous—roughly speaking, they represent types "less than proportionally."

Theorem 2b shows that utility-maximizing recommendations are generally diverse. More specifically, however, we show that when there is no decay in item quality, representation of a type varies *inversely* with the quality of items within that type. This holds empirically whenever the rate of decay is small. While perhaps counterintuitive, this is explained by the need to recommend more items from such a type to ensure that the user likes at least one. We isolate this case in Corollary 3, which we call the "milk and ice cream theorem," since it helps explains the paradoxical empirical fact that while consumers are more likely to buy milk, grocery stores devote much more aisle space to ice cream.

When the rate of decay is "severe," Theorems 2a and 2b collectively show that accuracy- and utility-maximizing recommendations coincide, and are diverse.

*Implications.* Our results lay out the specific ways in which diversity supports user utility, and thus inform how diversity should be incorporated into recommender systems. In particular:

- Maximizing user utility—properly conceived as incorporating consumption constraints—is often aligned with showing users a diverse set of items. Thus, to the extent that real-world systems do not show diverse recommendation sets, our results suggest that they are also failing to optimize user utility. Notably, this is true even before considering other ways in which diversity factors into utility (e.g., an intrinsic preference for diversity).
- Our results suggest principled approaches to diversify recommendations in a way that also optimizes utility. In particular, our results show that when users have consumption constraints, the relative likelihood a user prefers a specific type of item does not asymptotically affect the optimal representation of that item. Therefore, systems should recommend items relatively equally from a user's possible set of interests—even the niche interests.
- When the platform can estimate quality within a type (how often consumers like a specific ice cream flavor, conditional on wanting ice cream), the platform should recommend *more* items from types in which individual items are *less likely* to be satisfactory.

*Paper Outline.* In Section 2, we introduce our model. In Section 3, we introduce our theoretical results, first in a basic setting (Section 3.1) and then in a general setting (Section 3.2). In Section 4, we give an overview of our proof technique, sketching how we are able to derive our asymptotic results. In Section 5, we test our theoretical predictions in a range of computational experiments. In particular, we conduct a semi-synthetic experiment in which items and user preferences lie in a continuous space as estimated via matrix factorization, relaxing the assumption that there are a finite number of item and preference types. In Section 6, we conclude. Full proofs and an extended related work are left to the appendix.

## 2 MODEL

### 2.1 Specifying a recommendation setting

A recommender is tasked with recommending a fixed number of items to a user. There are $m$ types of items, and each item belongs to exactly one type. At recommendation time, a user prefers exactly one of these $m$ types of items. In our exposition, we will treat $m$ as fixed and omit notation that depends on $m$.

Types are indexed by $[m] = \{1, 2, \cdots, m\}$ and we let a user's type preference be given by a random variable $T$, such that $\Pr[T = t] = p_t$ (so that $\sum_{t=1}^{m} p_t = 1$). When a user prefers type $t$ (i.e., $T = t$), they only like items of type $t$. We assume that there are an arbitrarily large number items of each type, and that conditional on $T = t$, a user likes the $i$-th item of type $t$ independently with probability $q_{t,i}$. Without loss of generality, we let $q_{t,1} \geq q_{t,2} \geq \cdots$.

Formally, we let the random variable $V_{t,i}$ indicate if the user likes the $i$-th item of type $t$, so that

$$\Pr[V_{t,i} = 1] = \Pr[T = t]\Pr[V_{t,i} \mid T = t] \tag{1}$$

$$= p_t q_{t,i}. \tag{2}$$

Note that the random variables $V_{t,i}$ are independent conditional on $T$. A **recommendation setting** is thus characterized by:

(1) **type probabilities** $p_1, p_2, \cdots, p_t$;

(2) **conditional item probabilities** $q_{t,1}, q_{t,2}, \cdots$ for $t \in [m]$.

In what follows, we assume that a recommendation setting is specified, and will omit dependencies of certain quantities on the recommendation setting.

(Remark: Under standard measures of accuracy, like we consider here, items have binary value. However, it is possible to consider a setup in which values can be distributed according to arbitrary distributions. We provide such a setup in Appendix C, and give the analog of Theorem 1 in this setting.)

### 2.2 Choosing a set of recommendations

We now focus on the task of selecting $n$ items to recommend. In this case, a set of recommendations can be identified by an ordered tuple $S = (n_1, n_2, \cdots, n_m)$, where the recommender recommends the $i$-th item of type $t$ for all $t \in [m]$ and $i \in [n_t]$. In other words, $S$ represents the set of recommendations with the top $n_t$ items from each type.[1] We will let $\mathcal{S}_n := \left\{(n_1, n_2, \cdots, n_m) \in \mathbb{Z}_{\geq 0}^m : \sum_{t=1}^{m} n_t = n\right\}$ denote the set of recommendation sets of size $n$.

We consider two objectives by which to optimize a set of recommendations $S = (n_1, n_2, \cdots, n_m) \in \mathcal{S}_n$:

- **Accuracy**: The expected proportion of items in $S$ that the user likes, given by

$$\mathrm{acc}(S) := \mathbb{E}\left[\frac{1}{n}\sum_{t=1}^{m}\sum_{i=1}^{n_t} V_{t,i}\right] \tag{3}$$

$$= \frac{1}{n}\sum_{t=1}^{m} p_t \sum_{i=1}^{n_t} q_{t,i}. \tag{4}$$

acc is the standard notion of accuracy commonly used to evaluate machine learning models. By the linearity of expectation, it

is maximized by selecting the items with the highest $\mathbb{E}[V_{t,i}] = p_t q_{t,i}$—i.e., the individual items the user is most likely to like.

- **Utility** (w/ unit consumption constraint): The probability that a user likes at least one item in $S$, given by

$$\mathrm{util}_1(S) := \Pr\left[V_{t,i} = 1 \text{ for some } t \in [m], i \in [n_t]\right] \tag{5}$$

$$= 1 - \sum_{t=1}^{m} p_t \prod_{i=1}^{n_t}(1 - q_{t,i}). \tag{6}$$

$\mathrm{util}_1$ aligns with a user's satisfaction when they only intend to use one of the recommended items, as is common. In this case—e.g., when the goal is to find one restaurant to dine at, one movie to watch, or one website to visit—what matters is if the user likes at least one recommended item.

In our analysis, we characterize the accuracy- and utility-maximizing recommendation sets, given by the following notation.

**Definition 1** ($S_n$ and $S_{n,1}$). Given a specified recommendation setting, we let $S_n$ and $S_{n,1}$ denote the recommendation sets of size $n$ that maximize acc and $\mathrm{util}_1$, respectively:[2]

$$S_n := \underset{S \in \mathcal{S}_n}{\arg\max}\, \mathrm{acc}(S) \tag{7}$$

$$S_{n,1} := \underset{S \in \mathcal{S}_n}{\arg\max}\, \mathrm{util}_1(S). \tag{8}$$

To understand the diversity of $S_n$ and $S_{n,1}$, we consider how well-represented items of each type are.

**Definition 2** (Representation). For $S = (n_1, n_2, \cdots, n_m)$, define

$$r_t(S) = \frac{n_t}{\sum_{u=1}^{m} n_u}, \tag{9}$$

the *representation* of type $t$ in $S$.

Intuitively, sets with relatively equal representation across types are more diverse. In the following section, we will characterize—in terms of the type probabilities and conditional item probabilities—$r_t(S_n)$ and $r_t(S_{n,1})$ across several regimes.

## 3 RESULTS

We now introduce our theoretical results, which characterize the composition of the accuracy- and utility-maximizing sets $S_n$ and $S_{n,1}$. We begin in Section 3.1 by considering a basic setting that starkly contrasts the objectives acc and $\mathrm{util}_1$; the first has a strong trade-off with diversity, while the second is entirely aligned with diversity. In Section 3.2, we characterize the representation of item types, $r_t(S_n)$ and $r_t(S_{n,1})$, in a significantly more general setting, where we focus on the effects of different properties of the recommendation setting (i.e., providing comparative statics).

### 3.1 A Basic Setting

We start with a simple case of our model where we let the type probabilities $p_1, p_2, \cdots, p_m$ vary but hold the conditional item probabilities $q_{t,i} = q$ fixed for some $q \in (0, 1)$. This setting provides a clear illustration of the drastic effect incorporating a consumption constraint can have.

---

[1]In what follows, it will be clear that the recommender should only recommend the top items from each type. For example, the recommender would never recommend the first, second, and fourth item of a type, but not the third.

[2]It is sometimes possible for multiple sets of recommendations to maximize these objectives. In this case, our results hold when selecting any of these sets.

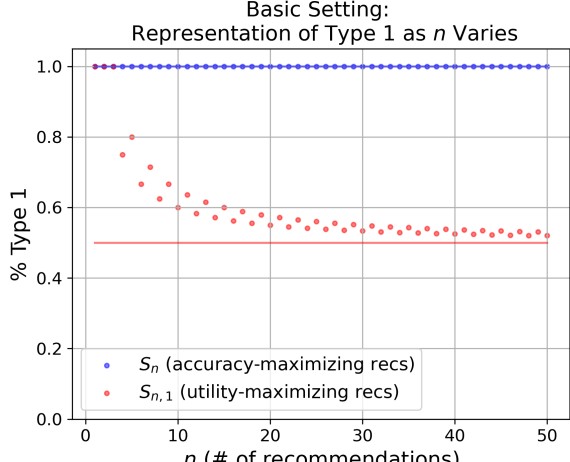

Basic Setting:
Representation of Type 1 as $n$ Varies

**Details:** A recommendation setting with $(p_1, p_2) = (0.8, 0.2)$, $q_{t,i} = 0.5$ **for all** $t, i$. **We plot empirical results (dots) and asymptotic theoretical results (solid lines) from Theorem 1.**

**Figure 1: An illustration of Theorem 1. Even while a user prefers type** 1 **much more often than type** 2, **as the number of recommendations increases, utility-maximizing recommendations (red) represent both types roughly equally. Meanwhile, accuracy-maximizing recommendations (blue) remain fully homogeneous throughout.**

To provide a concrete backdrop, suppose that there are $m$ types of movie genres. Then the probability a user is in the mood for genre $t$ is $p_t$. These type probabilities $p_t$ can vary, so that a user is more likely to be in the mood for some genres than others. Conditional on a user being in the mood for any genre $t$, any movie in that genre is liked by the user independently with probability $q$.

THEOREM 1. *Given type probabilities $p_1, p_2, \cdots, p_m$ and conditional item probabilities $q_{t,i} = q$,*

$$r_t(S_n) = \mathbb{1}_{\{t=\arg\max_t p_t\}} \tag{10}$$

$$\lim_{n \to \infty} r_t(S_{n,1}) = \frac{1}{m}. \tag{11}$$

This result conveys a strong dichotomy. (10) says that recommendations maximizing acc contains exclusively items from the genre the user is most likely to prefer, $\arg\max_t p_t$. This reflects the empirical existence of an accuracy-diversity trade-off: the accuracy-maximizing set of recommendations is fully homogeneous.

Meanwhile, (11) says that the set of recommendations that maximizes utility with a unit consumption constraint is fully diverse. Specifically, as the number of recommended items $n$ grows large, the recommender should recommend an equal proportion of items from each genre.[3] In this way, accounting for a user's consumption constraint dissolves the apparent accuracy-diversity trade-off; maximizing the probability a user likes *at least one* recommended movie is fully aligned with recommending a diverse set of movies.

---

[3]In fact, one can show that $r_t(S_{n,1}) = \frac{1}{m} + O\left(\frac{1}{n}\right)$, giving a relatively fast rate of convergence.

We now take a moment to convey the intuition behind Theorem 1. $S_n$ maximizes acc, which is equivalent to maximizing the expected number of recommended items the user likes. By linearity of expectation, this is achieved when recommending the individual items the user likes with the highest probability. The probability a user likes any item in genre $t$ is equal to $p_t q$. Therefore, the recommender should only recommend items from genre $\arg\max_t p_t$.

When the objective is to instead maximize util$_1$, the probability a user likes at least one item, recommending items from only one type is suboptimal. After recommending, say, many action movies, the probability the user is in the mood for action but does not like *any* of the recommended action movies is small ($p_t(1-q)^{n_t}$), where $n_t$ is the number of recommended action movies). At this point, recommending more action movies has diminishing returns, and one should hedge for the possibility that the user is in the mood for a different genre.

A surprising insight of Theorem 1 is that the type probabilities $p_t$ do not play any role asymptotically for $S_{n,1}$; even when a user watches more action than romance, the optimal set of recommendations represents the genres equally. To give some intuition, let $X$ be the event that a user does not like *any* recommended item. For an optimal set of recommendations $S$, $\Pr[X \mid T = t] = p_t q^{n_t}$ should equalized across $t$; otherwise, there would be an incentive to recommend more items from a type where this probability is higher. If $p_1 > p_2$, $\Pr[X \mid T = 1] = \Pr[X \mid T = 2]$ when recommending only a constant number $\log_q \frac{p_1}{p_2}$ more items from type 1 than type 2. So asymptotically, the proportion of items recommended from each type is equal. Representation thus quickly converges to this asymptotic value as $n$ increases; this is illustrated in one case in Figure 1.

### 3.2 A General Setting

We turn to a more general case where we consider heterogeneous conditional item probabilities and analyze comparative statics. Again, we consider arbitrary type probabilities $p_1, p_2, \cdots, p_m$. Now, we parameterize conditional item probabilities in the following way:

$$q_{t,i} := q_t(i + \beta)^{-\alpha}, \tag{12}$$

for $\alpha, \beta \geq 0$. This parameterization models heterogeneity of conditional item probabilities both within and across types. Some comments on the parameters $\alpha$, $\beta$, and $(q_1, q_2, \cdots, q_t)$:

- $\alpha$ is the *rate of decay* of item quality within a type. When $\alpha > 1$, this rate is extreme in the following sense: even if a user were recommended an infinite number of items in their preferred type, they (1) would only like a constant number of the items in expectation, and (2) with positive probability, would not like *any* of the recommended items.[4] Therefore, when the recommender has a reasonable "supply" of items, $\alpha \leq 1$ is realistic.
- $\beta$ parameterizes the initial steepness, with higher $\beta$ corresponding to lower initial steepness. $\beta$ does not end up appearing in our estimates.
- $q_1, q_2, \cdots, q_m$ are the *relative type qualities*. If $q_t > q_{t'}$, then $q_{t,i} > q_{t',i}$ for all $i$. Users can be less likely to like items of a certain type, even conditioned on preferring that type. This has

---

[4]Both facts boil down to the convergence of $\sum_{i=1}^{\infty} i^{-\alpha}$ when $\alpha > 1$.

**Table 1: Key notation in our model**

| | |
|---|---|
| $p_t$ | a *type probability*; a user prefers type $t$ with probability $p_t$ |
| $q_{t,i}$ | a *conditional item probability*; conditional on preferring type $t$, a user likes the $i$-th item of type $t$ with probability $q_{t,i}$; in our general setting, we parameterize $q_{t,i}$ as $q_{t,i} = q_t(i+\beta)^{-\alpha}$ |
| $\alpha$ | the *rate of decay* of item quality within a type |
| $q_t$ | a *relative type quality*; $q_t$ determines the relative quality of items in $t$ compared to other types |
| $S_n$ | the set of $n$ recommendations that maximizes acc, the expected proportion of items a user likes |
| $S_{n,1}$ | the set of $n$ recommendations that maximizes $\text{util}_1$, the probability a user likes at least one item |
| $r_t(S)$ | the proportion of items in $S$ of type $t$ |

two equivalent interpretations: the user is more picky when they prefer type $t'$, or the recommender has lower quality or more niche items in type $t'$.

We now give two main results, Theorem 2.A and Theorem 2.B which characterize $r_t(S_n)$ and $r_t(S_{n,1})$ in terms of these parameters.

Theorem 2.A (Accuracy-maximizing recommendations). *Given type probabilities $p_1, p_2, \cdots, p_m$ and conditional item probabilities $q_{t,i} := q_t(i+\beta)^{-\alpha}$,*

$$\lim_{n\to\infty} r_t(S_n) = \frac{(p_tq_t)^{1/\alpha}}{\sum_{u=1}^m (p_uq_u)^{1/\alpha}} \tag{13}$$

The key takeaway from Theorem 2.A is that accuracy-maximizing recommendations are more diverse when $\alpha$ is larger, i.e., when the quality of items in a type decays faster. Intuitively, this means that a recommender quickly runs out of high-quality items in a type, and thus benefits more from recommending items from other types. On the other hand, with small $\alpha$, the recommender has access to many high-quality items within each type.

Let us consider three cases of Theorem 1 to understand the functional form in (24):

- As $\alpha \to 0$, and if $\arg\max_t p_tq_t$ is unique,[5]
$$\lim_{n\to\infty} r_t(S_n) \to \mathbb{1}_{\{t=\arg\max_t p_tq_t\}}, \tag{14}$$
meaning that only the type with highest type probability is recommended.

- For $\alpha = 1$,
$$\lim_{n\to\infty} r_t(S_n) \propto p_tq_t, \tag{15}$$
meaning that a type is recommended in proportion to the probability a user likes items in that type.

- For $\alpha \to \infty$,
$$\lim_{n\to\infty} r_t(S_n) \to \frac{1}{m}, \tag{16}$$
meaning that an equal proportion of items from each type are recommended.

---

[5]If there are multiple types with the maximum type probability, one may check that in the limit, an equal proportion of items are recommended from these types, and a zero proportion from other types.

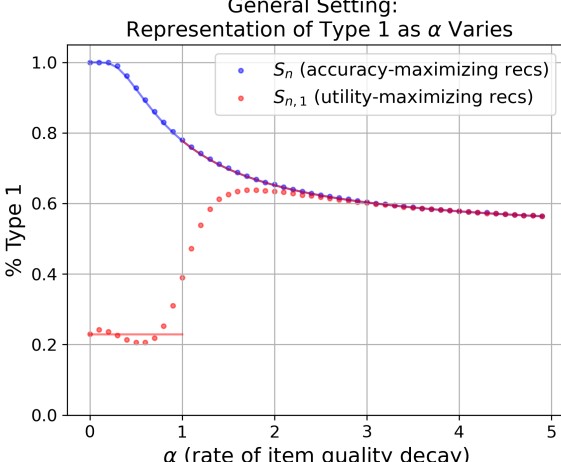

**Details: A** recommendation setting with $(p_1, p_2) = (0.6, 0.4)$, $(q_1, q_2) = (0.7, 0.3)$, $\beta = 2$, and $\alpha$ varying. Empirical results (dots) are for $n = 500$ and theoretical estimates (solid lines) are from Theorem 2.A and 2.B. For $\alpha \in [0, 1]$, the plotted theoretical estimate for $S_{n,1}$ are based on the theoretical result for $\alpha = 0$

**Figure 2: An illustration of Theorem 2.A and 2.B in a setting with two item types. When $\alpha < 1$, utility-maximizing recommendations (red) represent type 1 less than type 2 because 2 has lower relative conditional item probability, even though a user prefers 1 more often than 2. Behavior changes at $\alpha = 1$, after which accuracy- and utility-maximizing recommendations coincide.**

As $\alpha$ ranges from 0 to $\infty$, the amount of diversity in $S_n$ smoothly interpolates from maximal homogeneity to proportional representation to maximal diversity. Notably, when $\alpha \leq 1$, diversity is in the range between homogeneity and proportional representation. This suggests that in practice, the accuracy-diversity trade-off is particularly severe when the recommender has access to many high quality items of a type.

We next turn to utility-maximizing recommendations $S_{n,1}$, which account for a unit consumption constraint.

Theorem 2.B (Utility-maximizing recommendations). *Given type probabilities $p_1, p_2, \cdots, p_m$ and conditional item probabilities $q_{t,i} := q_t(i+\beta)^{-\alpha}$,*

$$\lim_{n\to\infty} r_t(S_{n,1}) = \begin{cases} \dfrac{\left(\log\frac{1}{1-q_t}\right)^{-1}}{\sum_{u=1}^m \left(\log\frac{1}{1-q_u}\right)^{-1}} & \alpha = 0 \\[4ex] \lim_{n\to\infty} r_t(S_{n,1}) = \dfrac{(p_tq_t)^{1/\alpha}}{\sum_{u=1}^m (p_uq_u)^{1/\alpha}} & \alpha > 1 \end{cases} . \tag{17}$$

The representation exhibits phase change at $\alpha = 1$. As mentioned in our discussion of the parameters, we expect $\alpha > 1$ represents an "extreme setting." We thus focus on the case $\alpha = 0$, which we

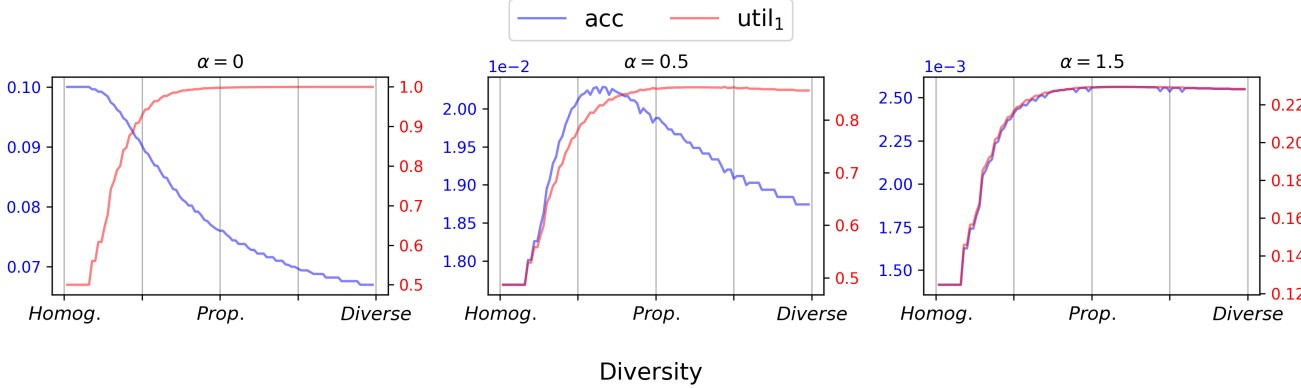

Details: Recommendation settings in which $(p_1, p_2, p_3) = (0.5, 0.3, 0.2)$ and $q_{t,i} = 0.2(i + 1)^{-\alpha}$ for $\alpha \in \{0, 0.5, 1.5\}$.

**Figure 3: In three settings with varying $\alpha$, we plot how accuracy and utility trade off with diversity. We consider sets $S$ with $n = 100$ items ranging from complete homogeneity (only type $1$ recommended), to proportional representation ($r_t(S) = p_t$) to complete diversity ($r_t(S) = \frac{1}{m}$), and plot $\mathrm{acc}(S)$ and $\mathrm{util}_1(S)$. Notice that in all of the plots, $\mathrm{util}_1$ is aligned with diversity, while $\mathrm{acc}(S)$ exhibits a strong trade-off with diversity when $\alpha = 0$, which becomes less severe for larger $\alpha$. These results agree with the predictions of Theorem 2.A and Theorem 2.B.**

pull out as its own result. (Note also that Theorem 1 is obtained by taking $q_1 = q_2 = \cdots = q_m$ and $\alpha = 0$.)

**Corollary 3 (The "milk and ice cream theorem").** *Given type probabilities $p_1, p_2, \cdots, p_m$ and conditional item probabilities $q_{t,i} = q_t(i + \beta)^{-\alpha}$, when $\alpha = 1$,*

$$\lim_{n \to \infty} r_t(S_{n,1}) = \frac{\left(\log \frac{1}{1 - q_t}\right)^{-1}}{\sum_{u=1}^{m} \left(\log \frac{1}{1 - q_u}\right)^{-1}}, \tag{18}$$

*meaning that $r_t(S_{n,1})$ is larger for types $t$ with lower $q_t$.*

The corollary's name references a paradoxical fact of grocery stores: that even while a customer is much more likely to buy milk than ice cream, significantly more aisle space is devoted to ice cream. The paradox can be resolved by the corollary in the following way. Let milk be type 1 and ice cream be type 2. A customer is more likely to purchase milk than ice cream, so $p_1 > p_2$. However, the probability a given carton of ice cream will satisfy a customer trying to purchase milk is lower than the probability that a given bottle of milk will satisfy a customer trying to purchase milk.[6] This means that $q_1 > q_2$. Corollary 3 reveals that more items should be recommended from the type with lower $q_t$—and, in fact, that $p_t$ is asymptotically irrelevant.

Corollary 3 demonstrates a broader insight into recommendations when the user has consumption constraints. Rather than focusing on type probabilities, it is more important to consider the conditional item values *within a type*—in particular, to recommend more items from types with low conditional item values. A good set of recommendations covers its bases across all possible preferences

---

[6]We note that this fact is complicated by customers' increasingly diversified tastes for—and the increased availability of—different types of plant-based milks [42].

of the user, and "covering" a type requires more items when items in that type have low conditional item probabilities.

Computational experiments suggest that behavior remains similar when $\alpha$ is small but larger than 0 (see Figure 2). However, Theorem 2.B shows that the behavior of $S_{n,1}$ changes when $\alpha > 1$. In fact, referring back to Theorem 2.A, we have that in this regime,

$$\lim_{n \to \infty} r_t(S_n) = \lim_{n \to \infty} r_t(S_{n,1}) = \frac{(p_t q_t)^{1/\alpha}}{\sum_{u=1}^{m} (p_u q_u)^{1/\alpha}}. \tag{19}$$

So when $\alpha > 1$, both $S_n$ and $S_{n,1}$ become more diverse as $\alpha$ increases. To provide rough intuition for this equality, note that for large $\alpha$, it is unlikely that a user will like more than one item in each type. Therefore, maximizing the likelihood a user likes at least one item ($\mathrm{util}_1$) is roughly equivalent to maximizing the total number of recommended items a user likes ($\mathrm{acc}$).

### 3.3 Summary of results

To summarize our theoretical results, while accuracy can exhibit a strong trade-off with diversity (especially when the rate of decay $\alpha$ is relatively small), our measure of utility that accounts for the capacity constraints of users does not exhibit a trade-off with diversity in the settings we study. This is exhibited in computational results shown in Figure 3, which shows how accuracy and utility vary as the level of diversity increases.

### 4 PROOF TECHNIQUE

We now provide an overview of our proof technique. We begin with some high-level intuition, from which our formal approach will deviate somewhat. The basic idea is that maximizing functions

of the form

$$\sum_{t=1}^{m} \lambda_t h(z_t), \tag{20}$$

subject to the constraint $(z_1, \cdots, z_m) \in \mathcal{S}^n$ is tractable when $f$ is simple (a monomial, for instance). For example, rough speaking, it is possible to solve

$$\lambda_1 h'(x_1) = \lambda_2 h'(x_2) = \cdots = \lambda_m h'(x_m), \tag{21}$$

and show that the integer-valued optimum must be near the real-valued optimum. While the objectives acc and $\mathtt{util}_1$ do not take the exact form as above, we show that there are choices of $\lambda_t$ such that the objectives evaluate to

$$\sum_{t=1}^{m} \lambda_t h_t(z_t), \tag{22}$$

where even while $h_t(z_t)$ may be complicated,

$$\lim_{z \to \infty} \frac{h_t(z)}{h(z)} = 1 \tag{23}$$

for a simple function $h$. We can then show that, under some reasonable assumptions on $h$, the solution to (23) is approximated by that of (20) in the limit as $n \to \infty$.

We now outline how this method can be used to prove Theorem 2.A when $\alpha < 1$ (the result is the same for $\alpha = 1$ and $\alpha > 1$, but these cases require separate analysis). Consider type probabilities $p_1, p_2, \cdots, p_m$ and conditional item probabilities $q_{t,i} := q_t (i + \beta)^{-\alpha}$. Then we would like to show that

$$\lim_{n \to \infty} r_t(S_n) = \frac{(p_t q_t)^{1/\alpha}}{\sum_{u=1}^{m} (p_u q_u)^{1/\alpha}} \tag{24}$$

Then $S_n = (z_1^{(n)}, z_2^{(n)}, \cdots, z_m^{(n)}) \in \mathcal{S}^n$ maximizes

$$\sum_{t=1}^{m} p_t \sum_{i=1}^{z_t} q_{t,i} = \sum_{t=1}^{m} p_t \sum_{i=1}^{z_t} q_t(i + \beta)^{-\alpha} = \sum_{t=1}^{m} \lambda_t h_t(z_t) \tag{25}$$

over $(z_1, z_2, \cdots, z_m) \in \mathcal{S}^n$, where

$$\lambda_t := \frac{p_t q_t}{1 - \alpha}, \qquad h_t(z) := (1 - \alpha) \sum_{i=1}^{z} (i + \beta)^{-\alpha}. \tag{26}$$

We then show that

$$\lim_{z \to \infty} \frac{h_t(z)}{h(z)} = 1 \tag{27}$$

where $h(z) = z^{1-\alpha}$. The result follows by using the following lemma, which is a subcase of Lemma A.1 in the appendix.

LEMMA 1. *Let $h(z) = z^\sigma$ for $\sigma \in (0, 1)$. For $t \in [m]$, suppose $h_t : \mathbb{Z}_{\geq 0} \to \mathbb{R}$ is monotonically increasing and strictly concave, and that*

$$\lim_{z \to \infty} \frac{h_t(z)}{h(z)} = 1. \tag{28}$$

*Let*

$$S^{(n)} = (z_1^{(n)}, z_2^{(n)}, \cdots, z_m^{(n)}) \in \arg\max_{(z_1, \cdots, z_m) \in \mathcal{S}^n} \sum_{t=1}^{m} \lambda_t h_t(z_t). \tag{29}$$

*Then*

$$\lim_{n \to \infty} r_t(S^{(n)}) = \frac{\lambda_t^{\frac{1}{1-\sigma}}}{\sum_{u=1}^{m} \lambda_u^{\frac{1}{1-\sigma}}}. \tag{30}$$

Taking $\sigma = 1 - \alpha$, we can apply the lemma to show that

$$\lim_{n \to \infty} r_t(S_n) = \frac{\left(\frac{p_t q_t}{1-\alpha}\right)^{1/\alpha}}{\sum_{u=1}^{m} \left(\frac{p_u q_u}{1-\alpha}\right)^{1/\alpha}} = \frac{(p_t q_t)^{1/\alpha}}{\sum_{u=1}^{m} (p_u q_u)^{1/\alpha}}. \tag{31}$$

# 5 COMPUTATIONAL EXPERIMENTS

We present results from a range of computational experiments, showing that our theoretical results generalize to practical settings.

## 5.1 Finite number of recommendations and beyond unit consumption constraints

We first focus on experiments in which $n$ is finite, ranging from small to moderate. We also relax the assumption that users have *unit* consumption constraints and consider varying rates of decay $\alpha < 1$. Consider the following more general version of utility corresponding to a consumption constraint of $k$:

$$\mathtt{util}_k(S) := \mathbb{E}\left[\max\left\{\sum_{t \in [m], i \in [n_t]} V_{t,i}, \quad k\right\}\right], \tag{32}$$

the value of the top $k$ items that a user likes. Accordingly, let

$$S_{n,k} := \arg\max_{S \in \mathcal{S}_n} \mathtt{util}_k(S), \tag{33}$$

where we recall that $\mathcal{S}_n$ is the set of all possible recommendation sets with $n$ items. Notice that our previous definitions of $\mathtt{util}_1$ and $S_{n,1}$ agree with this more general definition. (Indeed, when $k = 1$, (32) reduces to the probability $V_{t,i} = 1$ for at least one item.)

Also notice that $S_{n,n}$ maximizes the total number of items the user likes, and thus coincides with the accuracy-maximizing set $S_n$. We would expect our theoretical results about $\lim_{n \to \infty} r_t(S_{n,1})$ to be more accurate when $k$ is small, and to diverge from empirics when $k$ grows closer to $n$.

Here, we focus on a recommendation setting where there are two item types with $p_1 = 0.7$ and $p_2 = 0.3$. We let $q_{t,i} = q_t(i + \beta)^{-\alpha}$ where we fix $q_t = 0.5$ and $\beta = 1$, and only consider $\alpha < 1$. We focus on the case $\alpha < 1$ since it constitutes a reasonable rate of decay. We compare our empirical results to the estimate given by Theorem 2.B, which tells us that when $\alpha = 0$,

$$\lim_{n \to \infty} r_1(S_{n,1}) = \frac{1}{2}, \tag{34}$$

meaning that both types are equally represented. We evaluate how far empirical estimates of $r_1(S_{n,k})$ deviate from this prediction when $\alpha \in \{0, 0.2, 0.5, 0.9\}$ and $n$ and $k$ vary. Results are shown in Table 2. We observe that $r_1(S_{n,k})$ is near 0.5 whenever $\frac{k}{n}$ is relatively small and $\alpha$ is smaller. This suggests that our findings are robust when the consumption constraint $k$ is small in terms of $n$ (i.e., users only use a relatively small proportion of recommended items at a given time), and when the quality of items in a type does not decay significantly, i.e., there are many high-quality items per type.

Empirically, our results suggest that the theoretical estimate for $\alpha = 0$ is accurate for small $\alpha$, but begins to deteriorate as $\alpha$ approaches 1. (This is also corroborated by Figure 2, in which representation begins to change before reaching $\alpha = 1$.)

$$r_1(S_{n,k}) \text{ when } (p_1, p_2) = (0.7, 0.3) \text{ and } q_{t,i} = 0.5(i+1)^{-\alpha}.$$

| | $\alpha = 0$ | | | | $\alpha = 0.2$ | | | | $\alpha = 0.5$ | | | | $\alpha = 0.9$ | | | |
|---|---|---|---|---|---|---|---|---|---|---|---|---|---|---|---|---|
| $n$ | $k=1$ | $k=2$ | $k=5$ | $k=10$ | $k=1$ | $k=2$ | $k=5$ | $k=10$ | $k=1$ | $k=2$ | $k=5$ | $k=10$ | $k=1$ | $k=2$ | $k=5$ | $k=10$ |
| 10 | 0.6 | 0.6 | 1.0 | 1.0 | 0.6 | 0.6 | 1.0 | 1.0 | 0.6 | 0.7 | 0.9 | 0.9 | 0.7 | 0.7 | 0.8 | 0.8 |
| 20 | 0.55 | 0.55 | 0.6 | 1.0 | 0.55 | 0.6 | 0.7 | 1.0 | 0.6 | 0.65 | 0.8 | 0.95 | 0.65 | 0.7 | 0.75 | 0.8 |
| 50 | 0.52 | 0.48 | 0.52 | 0.54 | 0.52 | 0.56 | 0.54 | 0.66 | 0.56 | 0.58 | 0.64 | 0.86 | 0.64 | 0.62 | 0.78 | 0.76 |
| 100 | 0.5 | 0.49 | 0.49 | 0.49 | 0.52 | 0.49 | 0.49 | 0.53 | 0.55 | 0.56 | 0.57 | 0.76 | 0.63 | 0.62 | 0.7 | 0.78 |

Table 2: Empirical estimates of $r_1(S_{n,k})$ based on 5000 iterations for each setting (i.e., entry in table) for each possible set of recommendations. These can be compared with our theoretical result showing that $\lim_{n\to\infty} r_1(S_{n,1}) = 0.5$ for $\alpha = 0$. Blue indicates especially close to the result for $r_1(S_{n,k})$, and red indicates cases that deviate from the theoretical result. The result is most applicable when $k$ is small compared to $n$ and when $\alpha$ is smaller. Note also that $S_{n,k}$ is relatively diverse in all cases when $n$ is large compared to $k$, and with small $\alpha$ (i.e., there are many high-quality items per type).

We provide additional computational experiments in different settings in the appendix, as well as details for how we determine the empirically objective-maximizing recommendations.

## 5.2 Continuous Items and User Preferences

We now depart from our assumption that user preferences and items fall into discrete types, and instead represent both by embeddings on the unit $d$-dimensional sphere $S^d$. Given a user preference $t \in S^d$ and an item $v \in S^d$, we let the value of item $v$ be equal to the dot product $\max(t \cdot v, 0)$. Thus, items that are closer to a user's true preference have higher value and items cannot have negative value.

For a set of $n$ recommendations, we again let accuracy measure the average value of recommended items and utility measure the value of the best-recommended item (that of the highest value). We then compare the performance of an accuracy-maximizing set of recommendations with a heuristically constructed set of diverse recommendations. Our results are plotted in Figure 4. In alignment with our general findings, the accuracy-maximizing set of recommendations is homogeneous, while a diverse set of recommendations improves user utility.

Specifically, we use 10-dimensional embeddings trained using interaction data from GoodReads between 1000 users and 200 books. Embeddings are normalized to lie on $S^{10}$. We assume user preferences are drawn uniformly from the set of books they have interacted with (since these represent the range of interests the user has). We limit our experiments to the 206 users with at least 20 total interactions. For each user, we consider a "train set" of 10 of the user's past interactions. We then select $n$ of the 190 remaining books to recommend. The accuracy-maximizing set is chosen to maximize accuracy when user preferences are assumed to be drawn from the train set. The diverse set is chosen by choosing the closest items to each of the books in the train set (thus, covering the full range of user interests). We evaluate recommendations by randomly drawing a user preference from the books they have interacted with that were not already included in the train set. Additional details for our experiment are given in the appendix.

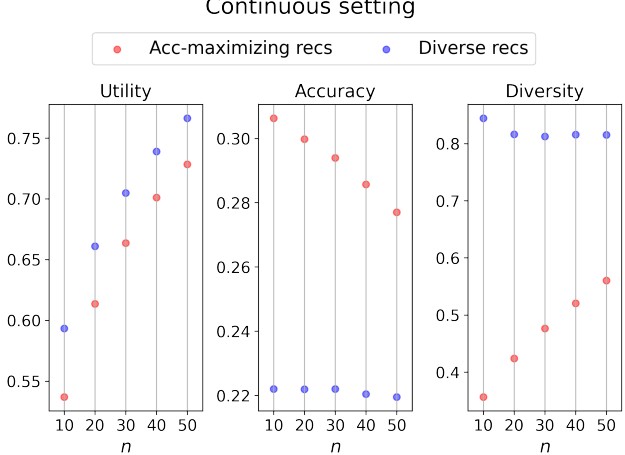

Figure 4: Book recommendations that maximize accuracy on a train set are more accurate in evaluation, but also achieve less utility and diversity as compared to a heuristically-chosen diverse set. Here, accuracy is the average value of recommended items, utility is the maximum value of recommended items, and diversity is the average cosine distance between recommendations. The numbers plotted are averages over 100 trials for each of the 206 users we evaluated.

## 6 CONCLUSION

We introduced and analyzed a model of recommendations that reconciles the apparent accuracy-diversity trade-off. In particular, we showed that accuracy is misaligned with user utility, because it does not consider the consumption constraints of users. By accounting for these consumption constraints, we found that user utility is in fact aligned with and supported by diversity. As a consequence, navigating the accuracy-diversity trade-off can be viewed as a way of incorporating diversity to help align accuracy with the more fundamental goal of user utility. Our results provide insight into how diversity can be incorporated in this manner.

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

## A  EXTENDED RELATED WORK

Our work sits at the intersection of two broad sets of work. On the one hand are arguments that diversity is key to achieving efficiency. On the other are those that cast diversity as in conflict with efficiency or accuracy, but perhaps that diversity should nevertheless be pursued as an axiomatic good.

Broadly, our work seeks to understand this tension by sharply characterizing the *amount* of diversity in efficient solutions, as a function of key setting characteristics: user utilities and consumption constraints, and uncertainty in the item quality distribution. In particular, our results characterize *in what settings* the intuition regarding diversity being efficient holds, and in what settings they may be in conflict.

*The (efficiency) benefits of diversity.* The importance of diversity for efficiency is an old idea present across many fields; Page [38] synthesizes the conceptual and empirical arguments in support of this principle. Hong and Page [24] develop a model in which a randomly selected team of problem solvers outperforms a team of the individually best-performing agents, due to diversity in problem solving perspective (Kleinberg and Raghu [30] show that, in some settings, there exist *tests* under which selecting the best-performing agents again becomes optimal). Kleinberg and Raghavan [29] show that constraints promoting diversity can improve efficiency when they work to counteract a decision-maker's biases. Agrawal et al. [3] develop an algorithm to diversify search results, to minimize the risk of user dissatisfaction. We are particularly influenced by the work of Steck [50], who presents the intuition that recommendations should be *calibrated*: "When a user has watched, say, 70 romance movies and 30 action movies, then it is reasonable to expect the personalized list of recommended movies to be comprised of about 70% romance and 30% action movies as well." Guo et al. [22] show that collaborative filtering-based recommendations may not be able to effectively show users such a diverse set of content, harming efficiency.

More broadly, researchers studying various combinatorial optimization problems may find it obvious that homogeneous solutions can be sub-optimal; indeed, in classical problems like *maximum coverage*, redundancy is undesirable.

Our work particularly is intimately connected to the large literature on assortment optimization [8, 11, 13–15, 18, 27, 32, 46, 47]. That literature also considers consumption-constrained consumer item selections based on an intermediary's recommendations (e.g., that customers picks one item according to a multinomial choice model). The literature primarily devises *approximation algorithms* to find the optimal recommendation ("assortment") as a function of the consumer's choice model, platform objective, and the item distribution. In other words, an implicit premise of this literature is that the naive approach of presenting the items with highest individual expected values is sub-optimal, i.e., that optimal assortments are not completely 'homogeneous.' On the other hand, optimal assortments are not necessarily diverse; roughly speaking, the results of El Housni and Topaloglu [15] imply that a standard assortment approach (Mixed MNL) might produce solutions that are not "diverse" enough to satisfy multiple customer types, and so there is benefit to personalize to each type.[7,8] Our work contributes to this literature by (a) examining the implicit premise that optimal assortments are not homogeneous (i.e., when is the naive[9] approach sufficient?); and (b) showing the characteristics under which optimal assortments are not diverse.

*Diversity and fairness as a contrast to efficiency and accuracy.* On the other hand, many works start with the premise that—although diversity may conflict with efficiency or accuracy—it is an axiomatic good that should be pursued. For example, diversity is often considered to be inherently desirable from a fairness perspective and user satisfaction perspective. As a result, there is a wide body of work devoted to optimizing for various metrics of diversity. A common approach (taken, for example, in Carbonell and Goldstein [10] and Gimpel et al. [19]) is to consider an objective function that balances a weighted measure of "accuracy" or "relevance" with a measure of diversity. More recently, Brown and Agarwal [9] consider set recommendation for an agent with adaptive preferences, to ensure that consumption over time is diverse. Numerous metrics for diversity have been proposed—we refer the reader to Kunaver and Požrl [33] for a survey. Similarly, the fair ranking and recommendation literature (see Patro et al. [40] and Zehlike et al. [53] for recent surveys) considers metrics and methods for fairness in such problems. On the other hand, empirical work has demonstrated that such tradeoffs may be small in practice [45]. Such formulations imply that there is a tension between diversity and measures of accuracy.

## B  DETAILS ON COMPUTATIONAL EXPERIMENTS / ADDITIONAL EXPERIMENTS

We provide additional details about the experiments we conduct in Section 5.

### B.1  Finite number of recommendations and beyond unit consumption constraints

We explain how we computed empirically optimal sets in 5.1. For each recommendation setting, we determine the set that maximizes $\text{util}_k$ by manually computing

$$\max\left\{\sum_{t\in[2], i\in[n_t]} V_{t,i}, \quad k\right\} \tag{35}$$

for sets with all possible combinations of item type representations. Specifically, in computing $S_{n,k}$, we consider the sets of the form $(i, n-i)$ for $i \in \{0, 1, \cdots, n\}$. For each of these sets $S$, we compute $\text{util}_k(S)$ directly, and take the average over 5000 iterations. We then choose the

---

[7]We thank the authors for highlighting this connection to us.

[8]Furthermore, as Chen et al. [11] recently characterize, standard assortment optimization approaches may be "unfair" to items in other ways.

[9]Note that *naive* is much simpler than the *greedy* approach studied in the literature, which picks items iteratively potentially as a function of previous items picked.

set with the maximum empirical expected utility, and display $r_1(S)$ in the Table 2. We consider additional settings, focusing on settings with varying $q_t$ and $\alpha > 1$, in Table 3 and Table 4 respectively.

$$r_1(S_{n,k}) \text{ when } (p_1, p_2) = (0.6, 0.4), (q_1, q_2) = (0.7, 0.3), \text{ and } q_{t,i} = q_t(i+1)^{-\alpha}.$$

| | $\alpha = 0$ | | | | $\alpha = 0.2$ | | | | $\alpha = 0.5$ | | | | $\alpha = 0.9$ | | | |
|---|---|---|---|---|---|---|---|---|---|---|---|---|---|---|---|---|
| $n$ | $k=1$ | $k=2$ | $k=5$ | $k=10$ | $k=1$ | $k=2$ | $k=5$ | $k=10$ | $k=1$ | $k=2$ | $k=5$ | $k=10$ | $k=1$ | $k=2$ | $k=5$ | $k=10$ |
| 10 | 0.3 | 0.4 | 0.8 | 1.0 | 0.4 | 0.5 | 1.0 | 1.0 | 0.5 | 0.6 | 1.0 | 1.0 | 0.6 | 0.8 | 0.9 | 0.9 |
| 20 | 0.3 | 0.3 | 0.4 | 0.8 | 0.3 | 0.4 | 0.55 | 1.0 | 0.4 | 0.5 | 0.8 | 1.0 | 0.55 | 0.7 | 0.8 | 0.8 |
| 50 | 0.26 | 0.18 | 0.28 | 0.34 | 0.26 | 0.3 | 0.32 | 0.5 | 0.3 | 0.38 | 0.56 | 0.88 | 0.44 | 0.58 | 0.72 | 0.8 |
| 100 | 0.24 | 0.11 | 0.16 | 0.26 | 0.25 | 0.19 | 0.25 | 0.34 | 0.26 | 0.29 | 0.39 | 0.68 | 0.39 | 0.43 | 0.73 | 0.85 |

**Table 3: Empirical estimates of $r_1(S_{n,k})$ based on 5000 iterations for each setting (i.e., entry in table) for each possible set of recommendations. The table illustrates the effect of the varying relative type qualities $q_1$ and $q_2$. Because $q_2$ is lower than $q_1$, we would expect type 1 to be represented less than type 2 according to our theoretical prediction from Corollary 3, which applies directly to the case $\alpha = 0$, $k = 1$, and $n \to \infty$. We highlight when this is the case in blue.**

$$r_1(S_{n,k}) \text{ when } (p_1, p_2) = (0.7, 0.3), q_{t,i} = 0.5(i+1)^{-\alpha}.$$

| | $\alpha = 1$ | | | | $\alpha = 1.5$ | | | | $\alpha = 2.0$ | | | | $\alpha = 2.5$ | | | |
|---|---|---|---|---|---|---|---|---|---|---|---|---|---|---|---|---|
| $n$ | $k=1$ | $k=2$ | $k=5$ | $k=10$ | $k=1$ | $k=2$ | $k=5$ | $k=10$ | $k=1$ | $k=2$ | $k=5$ | $k=10$ | $k=1$ | $k=2$ | $k=5$ | $k=10$ |
| 10 | 0.7 | 0.7 | 0.7 | 0.7 | 0.7 | 0.7 | 0.7 | 0.6 | 0.6 | 0.6 | 0.6 | 0.5 | 0.6 | 0.5 | 0.6 | 0.6 |
| 20 | 0.65 | 0.7 | 0.75 | 0.65 | 0.65 | 0.75 | 0.65 | 0.6 | 0.6 | 0.65 | 0.45 | 0.4 | 0.6 | 0.5 | 0.75 | 0.6 |
| 50 | 0.64 | 0.66 | 0.76 | 0.72 | 0.64 | 0.62 | 0.68 | 0.74 | 0.6 | 0.68 | 0.64 | 0.6 | 0.58 | 0.6 | 0.32 | 0.42 |
| 100 | 0.64 | 0.57 | 0.75 | 0.72 | 0.64 | 0.63 | 0.64 | 0.72 | 0.61 | 0.58 | 0.46 | 0.62 | 0.59 | 0.39 | 0.44 | 0.54 |

**Table 4: Empirical estimates of $r_1(S_{n,k})$ based on 5000 iterations for each setting (i.e., entry in table) for each possible set of recommendations. This table illustrates the regime in which $\alpha \geq 1$. In this regime, Theorem 2.A, which applies directly to the case $k = 1$, $\alpha > 1$, and $n \to \infty$, suggests that representation should be proportional to $p_t$ when $\alpha = 1$ representation should approach equal as $\alpha$ increases. Our empirical results appear to roughly reflect this trend for all pairs $(n, k)$.**

## B.2 Continuous Items and User Preferences

We provide details for our experiment in Section 5.2 on GoodReads data [51]. We used a subset of interaction data from 1000 users and 200 books, which we used to compute embeddings.[10] We considered the 206 users that had at least 20 book interactions.

Let $V$ be the set of all 200 book embeddings and $V_i$ be the set of books the user $i$ has interacted with. Suppose that $V_{\text{train}}$ is the train set of 10 book embeddings randomly drawn from $V_i$. Suppose that $V_i' = V_i \setminus V_{\text{train}}$ is the remaining set of embeddings.

Recall that a user's value of an embedding $v$ given their current preference $v_{\text{pref}}$ is given by

$$\max\{0, v \cdot v_{\text{pref}}\}. \tag{36}$$

Given a set of $n$ recommendations $S \subset S^d$, we evaluate it by drawing a random embedding $v_{\text{test}}$ from $V_i' \cap V_i$ as the user's current preference and considering the objectives

$$\text{acc}(S) = \frac{1}{n} \sum_{v \in S} u(v, v_{\text{test}}) \tag{37}$$

and

$$\text{util}_1(S) = \max_{v \in S} u(v, v_{\text{test}}), \tag{38}$$

the naturally analogs of acc and util$_1$ we considered in our theoretical results.

---

[10]The main dataset can be found at https://sites.google.com/eng.ucsd.edu/ucsdbookgraph/shelves. Embeddings were trained using the matrix factorization library libFM [44], which can be found at http://libfm.org/.

We constructed two sets of recommendations for each user given the training set $V_{\text{train}}$: the accuracy-maximizing set $S_n$ and a diverse set $S_{\text{diverse}}$. We construct these sets as follows.

$S_n$ is the set that maximizes average accuracy when user preferences are drawn from the train set:

$$\frac{1}{10} \sum_{v_{\text{pref}} \in V_{\text{train}}} \frac{1}{n} \sum_{v \in S_n} u(v, v_{\text{pref}}). \tag{39}$$

Computationally, we can determine this set $S$ by choosing the $n$ individual embeddings $v$ in $V'$ that maximize

$$\frac{1}{10} \sum_{v_{\text{pref}} \in V_{\text{train}}} u(v, v_{\text{pref}}). \tag{40}$$

We choose $S_{\text{diverse}}$ using a heuristic method. We iterate over items in the train set, and select the closest item to the train set in terms of cosine distance that has not yet been selected. At iteration $i \in [n]$, we let $v_{\text{pref}}$ be the $i \pmod{10}$-th item in $V_{\text{train}}$ and $S_{\text{diverse},i-1}$ be the set of $i - 1$ items selected so far. Then we construct $S_{\text{diverse},i}$ by adding the item in $V'$ that maximizes

$$v \cdot v_{\text{pref}} \tag{41}$$

to the set $S_{\text{diverse},i-1}$, where $S_{\text{diverse},0} = \emptyset$. We then choose $S_{\text{diverse}} = S_{\text{diverse},n}$.

To evaluate the diversity of a set $S$ of $n$ recommendations, we use the average cosine distance between embeddings:

$$\frac{1}{\binom{n}{2}} \sum_{v,v' \in S} 1 - v \cdot v'. \tag{42}$$

The results we report are averages over all users and over 100 independently drawn training sets for each user.

## C  A SETUP WITH ITEM VALUES FROM GENERAL DISTRIBUTIONS

In this section, we consider a model in which items can have values distributed arbitrarily over $\mathbb{R}$. In such a model, there are once again $m$ types of items indexed by $[m] = \{1, 2, \cdots, m\}$. A user prefers exactly one type of item, preferring type $t \in [m]$ with probability $p_t$. As before, $p_t$ give **type probabilities**.

Now, the value of the $i$-th item of type $t$ is a random variable $X_i^{(t)}$ if the user prefers type $t$ and 0 otherwise (so its expected value is $p_t \mathbb{E}[X_i^{(t)}]$). We refer to $X_i^{(t)}$ as a **conditional item value** (the value of an item conditional on the user preferring the item's type). Conditional item values are independent conditional on the user's preference. The case when $X_i^{(t)}$ is Bernoulli corresponds to our main setup.

Once specifying the type probabilities and conditional item values, we again define $S_n$ as the accuracy-maximizing set of recommendations, that which maximizes the expected total value of recommended items. We let $S_{n,k}$ be the set of recommendations that maximizes the expected value of the top $k$ recommended items, thus corresponding to a consumption constraint of $k$.

Recall that $r_t(S)$ gives the proportion of items in $S$ that are of type $t$. To succinctly introduce our main result in this setting—an analog to Theorem 1—we introduce the following way to measure diversity.

**Definition 3** ($\gamma$-homogeneity). A set $S$ is **$\gamma$-homogeneous** if for all $t \in [m]$,

$$r_t(S) = \frac{p_t^{\gamma}}{\sum_{i=1}^{m} p_i^{\gamma}}. \tag{43}$$

$\gamma$-homogeneity captures several intuitive notions of diversity, using $p_1, \cdots, p_m$ as a benchmark:
- When $\gamma = 0$, $r_t(S) = \frac{1}{m}$. There is "equal representation."
- When $\gamma = 1$, $r_t(S) = p_t$. There is "proportional representation," where an item type is represented in proportion to its likelihood.
- When $\gamma = \infty$, $r_t(S) = 1$ for $t = \arg\max_{i \in [m]} p_i$ and $r_t(S) = 0$ otherwise. There is "complete homogeneity," where only the highest-likelihood item type is represented.

A smaller $\gamma$ corresponds to more diversity, with $\gamma \leq 1$ indicating *at least proportional* representation. In practice, it is challenging to show that individual sets are $\gamma$-homogeneous; for one, since sets have an integer number of items from each type, it is typically impossible to obtain the exact ratios in (43). Instead, we will give primarily asymptotic results, showing that as $n$ grows large, the optimal set $S_{n,k}$ approaches $\gamma$-homogeneity. Formally, we define $\gamma$-homogeneity over sequences of sets:

**Definition 4** ($\gamma$-homogeneity for set sequences). A sequence of sets $\{S_n\}_{n=1}^{\infty}$ is **$\gamma$-homogeneous** if for all $t \in [m]$,

$$\lim_{n \to \infty} r_t(S_n) = \frac{p_t^{\gamma}}{\sum_{i=1}^{m} p_i^{\gamma}}. \tag{44}$$

We can then state our result as follows.

**THEOREM 4.** *Suppose $X_i^{(t)} \overset{\text{iid}}{\sim} \mathcal{D}$ where $\mathcal{D}$ has finite mean. Then the following statements hold.*

**Table 5: A summary of Theorem 4. For $X_i^{(t)} \overset{\text{iid}}{\sim} \mathcal{D}$, distributions $\mathcal{D}$ with heavier tails induce less diversity.**

| | bounded Thm. 1(ii) | | | exp. tail Thm. 1(iii) | heavy tail Thm. 1(iv) |
|---|---|---|---|---|---|
| example $\mathcal{D}$ | $\text{Beta}(\cdot, \beta)$ | | | $\text{Exp}(\lambda)$ | $\text{Pareto}(\alpha)$ |
| | $0 < \beta < 1$ | $\beta = 1$ | $\beta > 1$ | $\lambda > 0$ | $\alpha > 1$ |
| graph of pdf | | | | | |
| $\{S_{n,k}\}_{n=1}^{\infty}$ $\gamma$-homog. for $\gamma \in$ | $(0, 1/2)$ | $1/2$ | $(1/2, 1)$ | $1$ (i.e., proportional) | $(1, \infty)$ |

$\longleftarrow$ more diverse $\qquad$ less diverse $\longrightarrow$

(i) **[Finite Discrete]** If $\mathcal{D}$ is a finite discrete distribution, $\{S_{n,k}\}_{n=1}^{\infty}$ is 0-homogeneous.

(ii) **[Bounded]** If $\mathcal{D}$ has support bounded from above by $M$ with pdf $f_{\mathcal{D}}$ satisfying

$$\lim_{x \to M} \frac{f_{\mathcal{D}}(x)}{(M - x)^{\beta - 1}} = c \tag{45}$$

for some $\beta, c > 0$, then $\{S_{n,k}\}_{n=1}^{\infty}$ is $\frac{\beta}{\beta+1}$-homogeneous.

(This pdf class contains beta distributions, including the uniform distribution.)

(iii) **[Exponential tail]** If $\mathcal{D} = \text{Exp}(\lambda)$ for $\lambda > 0$, then $\{S_{n,k}\}_{n=1}^{\infty}$ is 1-homogeneous.

(iv) **[Heavy tail]** If $\mathcal{D} = \text{Pareto}(\alpha)$ for $\alpha > 1$, then $\{S_{n,k}\}_{n=1}^{\infty}$ is $\frac{\alpha}{\alpha-1}$-homogeneous.

Additionally,

(v) $S_n$ contains only items of type $t = \arg\max_{t \in [m]} p_t$.

As Table 5 illustrates, the theorem shows how for fixed $k$, the diversity of optimal solutions depends on the tail behavior of $\mathcal{D}$. In fact, we can obtain $\gamma$-homogeneity for any $\gamma$:

COROLLARY 5. *For any $\gamma \geq 0$, there exists $\mathcal{D}$ such that when $X_i^{(t)} \overset{\text{iid}}{\sim} \mathcal{D}$ and $k$ is fixed, $\{S_{n,k}\}_{n=1}^{\infty}$ is $\gamma$-homogeneous.*

Intuitively, heavy-tailed distributions (part (iv)) induce less diverse recommendations since the marginal returns of recommending more items from the same type remains high: drawing more samples from a heavy-tailed distribution produces ever-increasing item values. This contrasts with bounded distributions like the uniform distribution (part (ii)), where once an item has close to the maximum value, additional draws of that type will not further improve the utility significantly.

Part (i) of the theorem includes Theorem 1 as a subcase when $\mathcal{D}$ is Bernoulli. We prove Theorem 4 in appendix E.

# D MAIN PROOFS

## D.1 A central lemma: connecting diminishing returns to diversity

Let $\mathbb{Z}_{\geq 0}$ denote the set of non-negative integers and $z_n \subset \mathbb{Z}_{\geq 0}^m$ denote the set of $m$-tuples whose elements sum to $n$. We will say that a function $h : \mathbb{Z}_{\geq 0} \to \mathbb{R}$ is **strictly concave** if $h_t(z + 1) - h_t(z) < h_t(z) - h_t(z - 1)$ for all $a$.

**LEMMA D.1.** *Consider an integer $m$ and $p_1, p_2, \cdots, p_m \geq 0$. Let $h : \mathbb{Z}_{\geq 0} \to \mathbb{R}$ be monotonically increasing. For each positive integer $n$, choose $(z_1^{(n)}, \cdots, z_m^{(n)})$ such that*

$$(z_1^{(n)}, \cdots, z_m^{(n)}) \in \underset{(z_1, \cdots, z_m) \in z_n}{\arg\max} \sum_{t=1}^m \lambda_t h_t(z_t), \tag{46}$$

*and define*

$$r_t^{(n)} := \frac{z_t^{(n)}}{n}. \tag{47}$$

*Then the following statements hold.*

(i) *Suppose there exist constants $A, B > 0$ and $\sigma < 0$ such that*

$$\lim_{a \to \infty} \frac{\log(A - h_t(z))}{Bz^\sigma} = 1. \tag{48}$$

*Then*

$$\lim_{n \to \infty} r_t^{(n)} = \frac{1}{m}. \tag{49}$$

(ii) *Suppose there exist constants $A_1, A_2, \cdots, A_m, B > 0$ and $\sigma < 0$ such that*

$$\lim_{a \to \infty} \frac{A_t - h_t(z)}{Bz^\sigma} = 1. \tag{50}$$

*Then*

$$\lim_{n \to \infty} r_t^{(n)} = \frac{\lambda_t^{\frac{1}{1-\sigma}}}{\sum_{i=1}^m \lambda_i^{\frac{1}{1-\sigma}}}. \tag{51}$$

(iii) *Suppose $h$ is strictly concave, and that there exist constants $B, C > 0$ such that*

$$\lim_{a \to \infty} h_t(z) - B \log a - C = 0. \tag{52}$$

*Then*

$$\lim_{n \to \infty} r_t^{(n)} = \lambda_t. \tag{53}$$

(iv) *Suppose $h$ is strictly concave, and that there exist constants $B > 0$ and $0 < \sigma < 1$ such that*

$$\lim_{a \to \infty} \frac{h_t(z)}{Bz^\sigma} = 1. \tag{54}$$

*Then*

$$\lim_{n \to \infty} r_t^{(n)} = \frac{\lambda_t^{\frac{1}{1-\sigma}}}{\sum_{i=1}^m \lambda_i^{\frac{1}{1-\sigma}}}. \tag{55}$$

We spend the remainder of the section proving Lemma D.1. A useful first step is to show that in each of parts (i)-(iv), we have that

$$\lim_{n \to \infty} z_t^{(n)} = \infty \tag{56}$$

for each $t \in [m]$, allowing us to use the asymptotic assumptions in the lemma's statement.

Assume for the sake of contradiction that there exists $t \in [m]$ and an integer $d$ such that for any integer $N$ there exists $n > N$ for which $z_t^{(n)} < d$. Since $h$ is strictly increasing and $d$ is finite, there exists $\delta > 0$ such that

$$h_t(z + 1) - h_t(z) > \delta \tag{57}$$

for all $a < d$. Also, there exists an integer $N'$ such that for all $a > N'$,

$$h_t(z) - h_t(z - 1) < \delta \cdot \min_{i \in [m]} \frac{\lambda_t}{\lambda_i}. \tag{58}$$

(58) holds in parts (i)-(ii) because $h$ is monotonically increasing and is upper bounded by $A$, and in parts (iii)-(iv) because $h$ is strictly concave.

Now consider $N = N'm$. Then there exists $n > N$ such that $z_t^{(n)} < d$. Since $\sum_{t=1}^m z_t^{(n)} = n > N'm$, there exists $t' \in [m]$ such that $z_{t'}^{(n)} > N'$. Thus,

$$p_{t'} h_t(z_{t'}^{(n)}) - p_{t'} h_t(z_{t'}^{(n)} - 1) < \lambda_t \delta < \lambda_t h_t(z_t^{(n)} + 1) - \lambda_t h_t(z_t^{(n)}), \tag{59}$$

which implies that the switch $z_t^{(n)} \to z_t^{(n)} + 1, z_{t'}^{(n)} \to z_{t'}^{(n)} - 1$ increases

$$\sum_{t=1}^m \lambda_t h_t(z_t^{(n)}), \tag{60}$$

contradicting the optimality of $(z_1^{(n)}, \cdots, z_m^{(n)})$.

With (56) in hand, we turn to the bulk of the proof. In each part, we would like to show that

$$\lim_{n \to \infty} (r_1^{(n)}, \cdots, r_m^{(n)}) = (\widehat{r}_1, \cdots, \widehat{r}_m) \tag{61}$$

for some specified $(\widehat{r}_1, \cdots, \widehat{r}_m)$ depending on the part. We assume for the sake of contradiction that $\{(r_1^{(n)}, \cdots, r_m^{(n)})\}_{n=1}^{\infty}$ does not converge to $(\widehat{r}_1, \cdots, \widehat{r}_m)$. If this is the case, then by the Bolzano-Weierstrass theorem, since $[0,1]^m$ is compact, there is a subsequence

$$\{(r_1^{(s_i)}, \cdots, r_m^{(s_i)})\}_{i=1}^{\infty} \tag{62}$$

such that $\lim_{i \to \infty} (r_1^{(s_i)}, \cdots, r_m^{(s_i)}) = (r_1, \cdots, r_m)$ for some $(r_1, \cdots, r_m) \neq (\widehat{r}_1, \cdots, \widehat{r}_m)$. For notational ease, we will simply assume that

$$\lim_{n \to \infty} (r_1^{(n)}, \cdots, r_m^{(n)}) = (r_1, \cdots, r_m) \tag{63}$$

for some $(r_1, \cdots, r_m) \neq (\widehat{r}_1, \cdots, \widehat{r}_m)$. The proof holds analogously when the subsequence $\{(r_1^{(s_i)}, \cdots, r_m^{(s_i)})\}_{i=1}^{\infty}$ differs from $\{(r_1^{(n)}, \cdots, r_m^{(n)})\}_{n=1}^{\infty}$.

Then consider any sequence $\{(\widehat{z}_1^{(n)}, \cdots, \widehat{z}_m^{(n)})\}_{n=1}^{\infty}$ such that

$$\lim_{n \to \infty} \left( \frac{\widehat{z}_1^{(n)}}{n}, \cdots, \frac{\widehat{z}_m^{(n)}}{n} \right) = (\widehat{r}_1, \cdots, \widehat{r}_m). \tag{64}$$

(Clearly, such a sequence exists.) In each part, we will show that for sufficiently large $n$,

$$\sum_{t=1}^{m} \lambda_t h_t(z_t^{(n)}) < \sum_{t=1}^{m} \lambda_t h_t(\widehat{z}_t^{(n)}), \tag{65}$$

contradicting the optimality of $(z_1^{(n)}, \cdots, z_m^{(n)})$. To complete the proof, we analyze each part separately:

(i) In this part, there exist constants $A, B > 0$ and $\sigma < 0$ such that

$$\lim_{a \to \infty} \frac{\log(A - h_t(z))}{Bz^\sigma} = 1. \tag{66}$$

We set $\widehat{r}_t := \frac{1}{m}$ for each $t \in [m]$. Observe that

$$\lim_{a \to \infty} \frac{\log(A - h_t(z))}{Bz^\sigma} = 1 \tag{67}$$

implies that for all $\epsilon > 0$, there exists $c$ such that for all $a > c$,

$$e^{(1-\epsilon)Bz^\sigma} \leq A - h_t(z) \leq e^{(1+\epsilon)Bz^\sigma}. \tag{68}$$

Then observe that by taking sufficiently small $\epsilon$, we have

$$\lim_{n \to \infty} \frac{\sum_{t=1}^{m} \lambda_t (A - h_t(z_t^{(n)}))}{\sum_{t=1}^{m} \lambda_t (A - h_t(\widehat{z}_t^{(n)}))} \geq \lim_{n \to \infty} \frac{\sum_{t=1}^{m} \lambda_t \exp[(1-\epsilon)B(z_t^{(n)})^\sigma]}{\sum_{t=1}^{m} \lambda_t \exp[(1+\epsilon)B(\widehat{z}_t^{(n)})^\sigma]} \tag{69}$$

$$= \lim_{n \to \infty} \frac{\sum_{t=1}^{m} \lambda_t \exp[(1-\epsilon)B(nr_t)^\sigma]}{\exp[(1+\epsilon)B(n/m)^\sigma]} \tag{70}$$

$$= \lim_{n \to \infty} \sum_{t=1}^{m} \lambda_t \exp[Bn^\sigma((1-\epsilon)r_t^\sigma - (1+\epsilon)(1/m)^\sigma)] \tag{71}$$

$$= \infty \tag{72}$$

where the last limit holds for $\epsilon$ sufficiently small because $r_t - \frac{1}{m} > 0$ for some $t$.

It follows that for $n$ sufficiently large, $\sum_{t=1}^{m} \lambda_t h_t(z_t^{(n)}) < \sum_{t=1}^{m} \lambda_t h_t(\widehat{z}_t^{(n)})$, as desired.

(ii) In this part, there exist constants $A_1, \cdots, A_m, B > 0$ and $\sigma < 0$ such that

$$\lim_{a \to \infty} \frac{A_t - h_t(z)}{Bz^\sigma} = 1. \tag{73}$$

We set

$$\widehat{r}_t := \frac{\lambda_t^{\frac{1}{1-\sigma}}}{\sum_{i=1}^{m} \lambda_i^{\frac{1}{1-\sigma}}} \tag{74}$$

for each $t \in [m]$. Then observe that

$$\lim_{n\to\infty} \frac{\sum_{t=1}^m \lambda_t (A_t - h_t(z_t^{(n)}))}{\sum_{t=1}^m \lambda_t (A_t - h_t(\widehat{z}_t^{(n)}))} = \lim_{n\to\infty} \frac{\sum_{t=1}^m \lambda_t (A_t - h_t(z_t^{(n)}))}{\sum_{t=1}^m \lambda_t (A_t - h_t(\widehat{z}_t^{(n)}))} \cdot \lim_{n\to\infty} \frac{\sum_{t=1}^m \lambda_t B(z_t^{(n)})^\sigma}{\sum_{t=1}^m \lambda_t (A_t - h_t(z_t^{(n)}))} \cdot \lim_{n\to\infty} \frac{\sum_{t=1}^m \lambda_t (A_t - h_t(\widehat{z}_t^{(n)}))}{\sum_{t=1}^m \lambda_t B(\widehat{z}_t^{(n)})^\sigma} \tag{75}$$

$$= \lim_{n\to\infty} \frac{\sum_{t=1}^m \lambda_t (A_t - h_t(z_t^{(n)}))}{\sum_{t=1}^m \lambda_t (A_t - h_t(\widehat{z}_t^{(n)}))} \cdot \frac{\sum_{t=1}^m \lambda_t B(z_t^{(n)})^\sigma}{\sum_{t=1}^m \lambda_t (A_t - h_t(z_t^{(n)}))} \cdot \frac{\sum_{t=1}^m \lambda_t (A_t - h_t(\widehat{z}_t^{(n)}))}{\sum_{t=1}^m \lambda_t B(\widehat{z}_t^{(n)})^\sigma} \tag{76}$$

$$= \lim_{n\to\infty} \frac{\sum_{t=1}^m \lambda_t B(z_t^{(n)})^\sigma}{\sum_{t=1}^m \lambda_t B(\widehat{z}_t^{(n)})^\sigma} \tag{77}$$

$$= \frac{\sum_{t=1}^m \lambda_t r_t^\sigma}{\sum_{t=1}^m \lambda_t \widehat{r}_t^\sigma} > 1, \tag{78}$$

where (75) follows from the latter two limits being equal to 1, (76) follows from the product rule for limits, and (78) follows from the observation that for $\sigma < 0$

$$\sum_{t=1}^m \lambda_t x_t^\sigma, \tag{79}$$

subject to the constraint $\sum_{t=1}^m x_t = 1$ for $x_t \geq 0$ has a unique minimum at $(x_1, \cdots, x_m) = (\widehat{r}_1, \cdots, \widehat{r}_m)$. This is direct, for example, by using Lagrange multipliers. (78) implies that

$$\lim_{n\to\infty} \frac{\sum_{t=1}^m \lambda_t h_t(z_t^{(n)})}{\sum_{t=1}^m \lambda_t h_t(\widehat{z}_t^{(n)})} < 1. \tag{80}$$

It follows that for $n$ sufficiently large, $\sum_{t=1}^m \lambda_t h_t(z_t^{(n)}) < \sum_{t=1}^m \lambda_t h_t(\widehat{z}_t^{(n)})$, as desired.

(iii) In this part, $h$ is strictly concave, and there exist constants $B, C > 0$ such that

$$\lim_{a\to\infty} h_t(z) - B \log a - C = 0. \tag{81}$$

We set $\widehat{r}_t := \lambda_t$ for each $t \in [m]$. Then observe that

$$\lim_{n\to\infty} \sum_{t=1}^m \lambda_t h_t(z_t^{(n)}) - \sum_{t=1}^m \lambda_t h_t(\widehat{z}_t^{(n)}) \tag{82}$$

$$= \lim_{n\to\infty} \sum_{t=1}^m \lambda_t B \log z_t^{(n)} - \sum_{t=1}^m \lambda_t B \log \widehat{z}_t^{(n)} \tag{83}$$

$$= B \log n + B \sum_{t=1}^m \lambda_t \log r_t - B \log n - B \sum_{t=1}^m \lambda_t \log \widehat{r}_t \tag{84}$$

$$< 0. \tag{85}$$

The final inequality here follows from the observation that

$$\sum_{t=1}^m \lambda_t \log x_t, \tag{86}$$

subject to the constraint $\sum_{t=1}^m x_t = 1$ for $x_t > 0$ has a unique minimum at $(x_1, \cdots, x_m) = (\widehat{r}_1, \cdots, \widehat{r}_m)$. This is direct, for example, by using Lagrange multipliers.

It follows that for $n$ sufficiently large, $\sum_{t=1}^m \lambda_t h_t(z_t^{(n)}) < \sum_{t=1}^m \lambda_t h_t(\widehat{z}_t^{(n)})$, as desired.

(iv) In this part, $h$ is strictly concave, and there exist constants $B > 0$ and $0 < \sigma < 1$ such that

$$\lim_{a\to\infty} \frac{h_t(z)}{B z^\sigma} = 1. \tag{87}$$

We set

$$\widehat{r}_t := \frac{\lambda_t^{\frac{1}{1-\sigma}}}{\sum_{i=1}^m \lambda_i^{\frac{1}{1-\sigma}}} \tag{88}$$

for each $t \in [m]$. Then observe that

$$\lim_{n\to\infty} \frac{\sum_{t=1}^m \lambda_t h_t(z_t^{(n)})}{\sum_{t=1}^m \lambda_t h_t(\widehat{z}_t^{(n)})} = \lim_{n\to\infty} \frac{\sum_{t=1}^m \lambda_t B(z_t^{(n)})^\sigma}{\sum_{t=1}^m \lambda_t B(\widehat{z}_t^{(n)})^\sigma} = \frac{\sum_{t=1}^m \lambda_t r_t^\sigma}{\sum_{t=1}^m \lambda_t \widehat{r}_t^\sigma} < 1. \tag{89}$$

The first equality is a consequence of the asymptotic assumption on $h$ and the product rule for limits (as in part (ii)). The final inequality here follows from the observation that for $\sigma > 0$,

$$\sum_{t=1}^{m} \lambda_t x_t^{\sigma}, \tag{90}$$

subject to the constraint $\sum_{t=1}^{m} x_t = 1$ for $x_t > 0$ has a unique maximum at $(x_1, \cdots, x_m) = (\widehat{r}_1, \cdots, \widehat{r}_m)$. This is direct, for example, by using Lagrange multipliers.

It follows that for $n$ sufficiently large, $\sum_{t=1}^{m} \lambda_t h_t(z_t^{(n)}) < \sum_{t=1}^{m} \lambda_t h_t(\widehat{z}_t^{(n)})$, as desired.

## D.2 Proof of Theorem 2.A

We prove Theorem 2.A, where we are interested in the accuracy-maximizing set of recommendations. For recommendations $S = (z_1, z_2, \cdots, z_m)$, we have that

$$\sum_{t=1}^{m} \lambda_t h_t(z_t), \tag{91}$$

where

$$\lambda_t = p_t q_t \tag{92}$$

$$h_t(z) = \sum_{i=1}^{a} (i + \beta)^{-\alpha}. \tag{93}$$

We consider three cases: $0 < \alpha < 1$, $\alpha = 1$, and $\alpha > 1$.

*Case 1:* $0 < \alpha < 1$. For $0 < \alpha < 1$, observe that

$$\sum_{i=1}^{a} (i + \beta)^{-\alpha} < \int_{d}^{z+\beta} x^{-\alpha} \, dx = \left[ \frac{1}{1 - \alpha} x^{1-\alpha} \right]_{d}^{z+\beta} \tag{94}$$

$$= \frac{1}{1 - \alpha} (z + \beta)^{1-\alpha} - \frac{1}{1 - \alpha} d^{1-\alpha} \tag{95}$$

and

$$\sum_{i=1}^{a} (i + \beta)^{-\alpha} > \int_{\beta+1}^{z+\beta+1} x^{-\alpha} \, dx = \left[ \frac{1}{1 - \alpha} x^{1-\alpha} \right]_{\beta+1}^{z+\beta+1} \tag{96}$$

$$= \frac{1}{1 - \alpha} (z + \beta + 1)^{1-\alpha} - \frac{1}{1 - \alpha} (\beta + 1)^{1-\alpha}. \tag{97}$$

It follows that

$$\lim_{a \to \infty} \frac{h_t(z)}{z^{1-\alpha}} = \frac{1}{1 - \alpha}, \tag{98}$$

and the result in this case follows by applying Lemma D.1(iv).

*Case 2:* $\alpha = 1$. Now for $\alpha = 1$, we have that

$$\sum_{i=1}^{a} (i + \beta)^{-\alpha} = c \sum_{i=\beta+1}^{z+\beta} \frac{1}{i}. \tag{99}$$

$$\lim_{a \to \infty} h_t(z) - c \log a + c\gamma - c \sum_{i=1}^{d} \frac{1}{i} = 0, \tag{100}$$

where $\gamma$ is the Euler-Mascheroni constant The result in this case follows by applying Lemma D.1(iii).

*Case 3:* $\alpha > 1$. Finally, for $\alpha > 1$, we have that

$$\sum_{i=1}^{\infty} (i + \beta)^{-\alpha} = S \tag{101}$$

for some finite $S$. Then note that

$$\sum_{i=1}^{a} (i + \beta)^{-\alpha} = S - \sum_{i=z+1}^{\infty} (i + \beta)^{-\alpha}. \tag{102}$$

Then we have

$$\sum_{i=z+1}^{\infty} (i+\beta)^{-\alpha} < \int_a^{\infty} x^{-\alpha}\,dx = \left[\frac{1}{1-\alpha}x^{1-\alpha}\right]_a^{\infty} = -\frac{1}{1-\alpha}z^{1-\alpha} \tag{103}$$

and

$$\sum_{i=z+1}^{\infty} (i+\beta)^{-\alpha} > \int_{z+1}^{\infty} x^{-\alpha}\,dx = \left[\frac{1}{1-\alpha}x^{1-\alpha}\right]_{z+1}^{\infty} = -\frac{1}{1-\alpha}(z+1)^{1-\alpha} \tag{104}$$

It follows that

$$\lim_{a\to\infty} \frac{h_t(z)}{S - \frac{1}{\alpha-1}z^{1-\alpha}} = 1. \tag{105}$$

The result in this case follows again by applying Lemma D.1(ii).

## D.3 Proof of Theorem 2.B

We are now considered in utility-maximizing recommendations, which maximize the probability that a user is satisfied with *at least one* recommendation.

*D.3.1 Case when $\alpha = 0$.* The following lemma is useful for showing that—for the class of problems we consider here—optimal integer solutions are well-approximated by optimal real solutions.

LEMMA D.2. *Let $g_1, g_2, \cdots, g_m : [0, \infty)^m \to \mathbb{R}$ be strictly convex functions over the non-negative reals. Then define*

$$g(x_1, \cdots, x_m) := \sum_{t=1}^{m} g_t(x_t). \tag{106}$$

*Then, under the constraint that $\sum_{t=1}^{m} x_t = n$, if $(x_1^*, \cdots, x_m^*)$ is the maximum of $g$ over the non-negative reals and $(z_1^*, \cdots, z_m^*)$ is the maximum of $g$ over the non-negative integers, then*

$$\lfloor x_t^* \rfloor - m < z_t^* < \lfloor x_t^* \rfloor + m \tag{107}$$

*for all $t$.*

PROOF. The key idea is to show that there cannot be $i, j$ such that $z_i^* \geq \lceil x_i \rceil + 1$ and $z_j^* \leq \lceil x_j \rceil - 1$. If such a pair does exist, we show that

$$g(\cdots, z_i^* - 1, \cdots, z_j^* + 1, \cdots) \geq g(\cdots, z_i^*, \cdots, z_j^*, \cdots), \tag{108}$$

contradicting the optimality of $z_1^*, \cdots, z_m^*$. It suffices to show that

$$g_i(z_i^* - 1) + g_j(z_j^* + 1) \geq g_i(z_i^*) + g_j(z_j^*), \tag{109}$$

or equivalently,

$$g_j(z_j^* + 1) - g_j(z_j^*) \geq g_i(z_i^*) - g_i(z_i^* - 1). \tag{110}$$

This holds, as

$$g_j(z_j^* + 1) - g_j(z_j^*) \geq \frac{\partial g_j}{\partial x_j} = \frac{\partial g_i}{\partial x_i} \geq g_i(z_i^*) - g_i(z_i^* - 1). \tag{111}$$

□

We now prove Theorem 2.B when $\alpha = 0$.

PROOF. Given recommendations $S = (z_1, z_2, \cdots, z_m)$ we have that,
We would like to find $z_1, \cdots, z_m$ that maximizes

$$\sum_{t=1}^{m} p_t \left(1 - (1-q_t)^{z_t}\right) = 1 - \sum_{t=1}^{m} p_t (1-q_t)^{z_t}. \tag{112}$$

subject to the constraint $\sum_{t=1}^{m} p_t = n$. This is equivalent to minimizing

$$\sum_{t=1}^{m} p_t (1-q_t)^{z_t}. \tag{113}$$

Now define a function

$$g : [0, \infty)^m \to \mathbb{R}, \quad (x_1, \cdots, x_m) \mapsto \sum_{t=1}^{m} p_t (1-q_t)^{x_t}. \tag{114}$$

Subject to the constraint $\sum_{t=1}^{m} x_t = n$, $g(x_1, \cdots, x_m)$ is maximized exactly when

$$\frac{\partial g}{\partial x_1} = \frac{\partial g}{\partial x_2} = \cdots = \frac{\partial g}{\partial x_m}. \tag{115}$$

We have

$$\frac{\partial g}{\partial x_t} = -p_t (1 - q_t)^{x_t} \log(1 - q_t). \tag{116}$$

Solving $\partial g / \partial x_i = \partial g / \partial x_j$ gives

$$p_i (1 - q_i)^{x_i} \log(1 - q_i) = p_j (1 - q_j)^{x_j} \log(1 - q_j) \tag{117}$$

$$\implies \log p_i + x_i \log(1 - q_i) + \log \log(1 - q_i) = \log p_j + x_j \log(1 - q_j) + \log \log(1 - q_j) \tag{118}$$

It follows that $z_t \propto \frac{1}{\log(1-q_t)}$ for all $t$, where we have applied Lemma D.2. □

*D.3.2 Case when $\alpha > 1$.* We now consider the case $\alpha > 1$. Given recommendations $S = (z_1, z_2, \cdots, z_m)$ we have that

$$\texttt{util}_1(S) = \sum_{t=1}^{m} \lambda_t h_t(z_t), \tag{119}$$

where we set

$$\lambda_t = p_t q_t \tag{120}$$

$$h_t(z) = \frac{1 - \prod_{i=1}^{z} (1 - q_t(i + \beta)^{-\alpha})}{q_t}. \tag{121}$$

It suffices now to show the desired asymptotic properties for $h$ depending on $\alpha$, and applying Lemma D.1.

Note that

$$\prod_{i=\beta+1}^{\infty} (1 - q_t i^{-\alpha}) = S_t \tag{122}$$

for a finite constant $S_t$.

We note the following fact, which will be helpful in our analysis:

$$1 - x > e^{-x - x^2} \quad \text{for } 0 < x < \frac{1}{2}. \tag{123}$$

We have that

$$\prod_{i=z+\beta+1}^{\infty} (1 - q_t i^{-\alpha}) < \prod_{i=z+\beta+1}^{\infty} e^{-q_t i^{-\alpha}} \tag{124}$$

$$= \exp\left[ \sum_{i=z+\beta+1}^{\infty} -q_t i^{-\alpha} \right] \tag{125}$$

$$< \exp\left[ \int_{z+\beta+1}^{\infty} -q_t x^{-\alpha} \, dx \right] \tag{126}$$

$$= \exp\left[ -\left[ \frac{q_t x^{1-\alpha}}{1 - \alpha} \right]_{z+\beta+1}^{\infty} \right] \tag{127}$$

$$= \exp\left[ -\frac{q_t}{1 - \alpha} (z + \beta + 1)^{1-\alpha} \right] \tag{128}$$

Therefore,

$$\prod_{i=1}^{z+\beta} (1 - q_t i^{-\alpha}) = \frac{S_t}{\prod_{i=z+\beta+1}^{\infty} (1 - q_t i^{-\alpha})} > S_t / \exp\left[ -\frac{q_t}{1 - \alpha} (z + \beta + 1)^{1-\alpha} \right]. \tag{129}$$

Also,

$$\prod_{i=z+\beta+1}^{\infty} (1 - q_t i^{-\alpha}) > \prod_{i=z+\beta+1}^{\infty} e^{-q_t i^{-\alpha} - q_t^2 i^{-2\alpha}} \tag{130}$$

$$= \exp\left[\sum_{i=z+\beta+1}^{\infty} -q_t i^{-\alpha} - q_t^2 i^{-2\alpha}\right] \tag{131}$$

$$< \exp\left[\int_{z+\beta}^{\infty} -q_t x^{-\alpha} - q_t^2 x^{-2\alpha} \, dx\right] \tag{132}$$

$$= \exp\left[-\left[\frac{q_t x^{1-\alpha}}{1-\alpha} + \frac{q_t^2 x^{2-\alpha}}{2-\alpha}\right]_{z+\beta}^{\infty}\right] \tag{133}$$

$$= \exp\left[-\frac{q_t}{1-\alpha}(z+\beta)^{1-\alpha} - \frac{q_t^2}{1-2\alpha}(z+\beta)^{2-\alpha}\right], \tag{134}$$

where the first inequality holds for $a$ sufficiently large due to (123).

Therefore,

$$\prod_{i=1}^{z+\beta}(1 - q_t i^{-\alpha}) = \frac{S_t}{\prod_{i=z+\beta+1}^{\infty}(1 - q_t i^{-\alpha})} < S_t / \exp\left[-\frac{q_t}{1-\alpha}(z+\beta-1)^{1-\alpha} - \frac{q_t^2}{1-2\alpha}(z+\beta)^{2-\alpha}\right]. \tag{135}$$

Now observe that

$$\lim_{a \to \infty} -\frac{q_t}{1-\alpha}(z+\beta+1)^{1-\alpha} = 0 \tag{136}$$

and

$$\lim_{a \to \infty} -\frac{q_t}{1-\alpha}(z+\beta-1)^{1-\alpha} - \frac{q_t^2}{1-2\alpha}(z+\beta)^{2-\alpha} = 0. \tag{137}$$

Therefore,

$$\lim_{a \to \infty} \frac{\prod_{i=1}^{z+\beta}(1-q_t i^{-\alpha})}{\frac{S_t}{1 - \frac{q_t}{1-\alpha}(z+\beta+1)^{1-\alpha}}} = 1. \tag{138}$$

Also note that

$$\frac{S_t}{1 - \frac{q_t}{1-\alpha}(z+\beta+1)^{1-\alpha}} = S_t - \frac{S_t \frac{q_t}{1-\alpha}}{(z+\beta+1)^{\alpha-1} + \frac{q_t}{1-\alpha}} \tag{139}$$

Also,

$$\lim_{a \to \infty} \frac{\prod_{i=1}^{z+\beta}(1-q_t i^{-\alpha})}{\frac{S_t}{1 - \frac{q_t}{1-\alpha}(z+\beta-1)^{1-\alpha} - \frac{q_t^2}{1-2\alpha}(z+\beta)^{2-\alpha}}} = 1 \tag{140}$$

and

$$\lim_{a \to \infty} \frac{1 - \frac{q_t}{1-\alpha}(z+\beta)^{1-\alpha} - \frac{q_t^2}{1-2\alpha}(z+\beta)^{2-\alpha}}{1 - \frac{q_t}{1-\alpha}(z+\beta)^{1-\alpha}} = 1. \tag{141}$$

It follows that

$$\lim_{a \to \infty} \frac{\frac{1-S_t}{q_t} - h_t(z)}{-\frac{S_t}{1-\alpha}z^{1-\alpha}} = 1 \tag{142}$$

as desired.

Taking $A_t = \frac{1-S_t}{q_t}$ and $B = -\frac{S_t}{1-\alpha}$, we have that

$$\lim_{a \to \infty} \frac{A_t - h_t(z)}{Bz^{1-\alpha}} = 1, \tag{143}$$

and the result follows from Lemma D.1(ii).

# E    PROOF OF THEOREM 4 (GENERAL DISTRIBUTIONS)

We now turn to the proofs of Theorem 4(i)-(iv). (Part (v) is clear.) In each of these parts, we consider a set of recommendations with $a_t$ items of type $t$ for each $t \in [m]$. Then observe that the expected total value of the $k$ highest value recommended items is equal to

$$\sum_{t=1}^{m} p_t h_t(a_t), \tag{144}$$

for

$$h : \mathbb{Z}_{\geq 0} \to \mathbb{R}, \quad h : a \mapsto \mathbb{E}\left[\text{top}_k\{X_1, \cdots, X_a \overset{\text{iid}}{\sim} \mathcal{D}\}\right], \tag{145}$$

where $\text{top}_k$ evaluates the sum of the $k$ highest values in a set. Intuitively, conditional on a user preferring type $t$, the top $k$ items are just the top $k$ items recommended of type $t$. The sum of their values, conditioned on the user preferring type $t$, is simply the sum of the $k$ highest values among $a$ random draws from $\mathcal{D}$. Clearly, $h$ here is monotonically increasing.

Then, with Lemma D.1 in hand, parts (i)-(iv) reduces to showing the following:

(i) If $\mathcal{D}$ is a finite discrete distribution, there exist constants $A, B > 0$ and $\sigma > 0$ such that

$$\lim_{a \to \infty} \frac{\log(A - h(a))}{Ba^{\sigma}} = 1. \tag{146}$$

(ii) If $\mathcal{D}$ has support bounded from above by $M$ with pdf $f_{\mathcal{D}}$ satisfying

$$\lim_{x \to M} \frac{f_{\mathcal{D}}(x)}{(M-x)^{\beta-1}} = c \tag{147}$$

for some $\beta, c > 0$, then there exist constants $A, B > 0$ such that

$$\lim_{a \to \infty} \frac{A - h(a)}{Ba^{-\frac{1}{\beta}}} = 1. \tag{148}$$

(iii) If $\mathcal{D} = \text{Exp}(\lambda)$ for $\lambda > 0$, then $h$ is strictly concave and there exists a constant $B > 0$ such that

$$\lim_{a \to \infty} \frac{h(a)}{B \log a} = 1. \tag{149}$$

(iv) If $\mathcal{D} = \text{Pareto}(\alpha)$ for $\alpha > 1$, then $h$ is strictly concave and there exists a constant $B > 0$ such that

$$\lim_{a \to \infty} \frac{h(a)}{Ba^{\frac{1}{\alpha}}} = 1. \tag{150}$$

The following identity, mentioned in ??, will be useful for parts (ii)-(iv).

PROPOSITION 6. For $X_i^{(t)} \overset{\text{iid}}{\sim} \mathcal{D}$,

$$h(a) = \sum_{i=1}^{\min\{k,a\}} \mu_{\mathcal{D}}(a - i + 1, a). \tag{151}$$

Recall that $\mu_{\mathcal{D}}(i, a)$ is the expected value of the $i$-th order statistic of $a$ random variables drawn i.i.d. from $\mathcal{D}$.

PROOF. Let $Y_{k,n}$ be the $k$-th order statistic of $n$ random variables distributed i.i.d. from $\mathcal{D}$. Then

$$\text{top}_k\{X_1^{(t)}, \cdots, X_a^{(t)}\} = \sum_{i=1}^{\min\{k,a\}} Y_{a-i+1,a}. \tag{152}$$

So, as desired,

$$\mathbb{E}\left[\text{top}_k\{X_1^{(t)}, \cdots, X_a^{(t)}\}\right] = \sum_{i=1}^{\min\{k,a\}} \mathbb{E}\left[Y_{a-i+1,a}\right] = \sum_{i=1}^{\min\{k,a\}} \mu_{\mathcal{D}}(a - i + 1, a), \tag{153}$$

where the first equality follows from (152) and the linearity of expectation.                                      □

*Proof of Theorem 4(i).* Suppose $\mathcal{D}$ has support $\{x_1, \cdots, x_r\}$ with $x_1 > \cdots > x_r$ such that for $X \sim \mathcal{D}$, $\Pr[X = x_1] = q$. Now consider a set of recommendations with $a_t$ items of type $t$ for each $t \in [m]$. Then consider $X_1, \cdots, X_a \stackrel{iid}{\sim} \mathcal{D}$. Let $E$ be the event that at least $k$ of $X_1, \cdots, X_a$ equal $x_1$. Then,

$$h(a) \geq \mathbb{E}[\text{top}_k\{X_1, \cdots, X_a\}|E] \cdot \Pr[E] \tag{154}$$

$$= x_1 k \cdot \left(1 - \sum_{j=0}^{k-1} \binom{a}{j}(1-q)^{a-j} q^j\right) \tag{155}$$

$$\geq x_1 k (1 - a^k (1-q)^{a-k+1}) \tag{156}$$

for all $a > 2$. Now let $E'$ be the event that at least one of $X_1, \cdots, X_a$ equals $x_1$. Then,

$$h(a) = \mathbb{E}[\text{top}_k\{X_1, \cdots, X_a\}|E'] \cdot \Pr[E'] + \mathbb{E}[\text{top}_k\{X_1, \cdots, X_a\}|\overline{E'}] \cdot (1 - \Pr[E']) \tag{157}$$

$$\leq x_1 k (1 - (1-q)^a) + x_2 k (1-q)^a \tag{158}$$

$$= x_1 k (1 - (1 - \frac{x_2}{x_1})(1-q)^a). \tag{159}$$

Now note that for $A = x_1 k$, we have that

$$x_1 k (1 - \frac{x_2}{x_1})(1-q)^a \leq A - h(a) \leq x_1 k a^k (1-q)^{a-k+1} \tag{160}$$

$$\log(x_1 k (1 - \frac{x_2}{x_1})(1-q)^a) \leq \log(A - h(a)) \leq \log(x_1 k a^k (1-q)^{a-k+1}) \tag{161}$$

$$\log(x_1 k) + \log(1 - \frac{x_2}{x_1}) + a\log(1-q) \leq \log(A - h(a)) \leq \log(x_1 k) + k\log(a) + (a - k + 1)\log(1-q). \tag{162}$$

It follows that for $B = \log(1 - q)$,

$$\lim_{a \to \infty} \frac{\log(A - h(a))}{Ba} = 1, \tag{163}$$

as desired. The result follows from Lemma D.1(i).

*Proof of Theorem 4(ii).* First recall from Proposition 6 that

$$h(a) = \sum_{i=1}^{\min\{k,a\}} \mu_{\mathcal{D}}(a - i + 1, a). \tag{164}$$

We will show that

$$\lim_{a \to \infty} \frac{Mk - h(a)}{Ba^{-\frac{1}{\beta}}} = 1 \tag{165}$$

for a constant $B > 0$. Theorem 4(ii) then follows immediately by applying Lemma D.1(ii) with $\sigma = -\frac{1}{\beta}$.

Consider a probability distribution $\mathcal{D}'$ with pdf $g_X(x) = f_X(M - x)$ and cdf $G_X(x)$. Then

$$\mu_{\mathcal{D}}(a - i + 1, a) = M - \mu_{\mathcal{D}'}(i, a), \tag{166}$$

which implies that

$$Mk - \sum_{i=1}^{k} \mu_{\mathcal{D}}(a - i + 1, a) = \sum_{i=1}^{k} \mu_{\mathcal{D}'}(i, a) \tag{167}$$

Since

$$\mu_{\mathcal{D}'}(i, a) = \sum_{j=0}^{i-1} \int_0^\infty \binom{a}{j} G_X(x)^j (1 - G_X(x))^{a-j} \, dx, \tag{168}$$

it remains to show that for all fixed $j$,

$$\lim_{a \to \infty} \frac{\int_0^\infty \binom{a}{j} G_X(x)^j (1 - G_X(x))^{a-j} \, dx}{a^{-\frac{1}{\beta}}} = B \tag{169}$$

for some constant $B$ (that can vary depending on $j$). Verifying (169) comprises the bulk of the technical work of the proof, and we isolate it in the following lemma.

LEMMA E.1. *For $\beta > 0$,*

$$\int_0^\infty \binom{a}{j} G_X(x)^j (1 - G_X(x))^{a-j} \, dx \propto a^{-\frac{1}{\beta}}. \tag{170}$$

Proof. We have that

$$\lim_{x \to 0^+} \frac{g_X(x)}{cx^{\beta-1}} = \lim_{x \to M^-} \frac{f_X(x)}{c(M-x)^{\beta-1}} = 1 \tag{171}$$

for a positive constant $c$. So for all $\epsilon > 0$ there exists $\delta > 0$ such that

$$(1-\epsilon)cx^{\beta-1} \le g_X(x) \le (1+\epsilon)cx^{\beta-1} \tag{172}$$

for all $x < \delta$. Now note that $g_X(x) \le (1+\epsilon)cx^{\beta-1}$ implies that

$$G_X(x) = \int_0^x g_X(u)\,du \le (1+\epsilon)\int_0^x cu^{\beta-1}\,du = (1+\epsilon)\frac{c}{\beta}x^\beta. \tag{173}$$

Likewise, $g_X(x) \ge (1-\epsilon)cx^{\beta-1}$ implies that

$$G_X(x) = \int_0^x g_X(u)\,du \ge (1-\epsilon)\int_0^x cu^{\beta-1}\,du = (1-\epsilon)\frac{c}{\beta}x^\beta. \tag{174}$$

Now write

$$a^{\frac{1}{\beta}} \int_0^\infty \binom{a}{j} G_X(x)^j (1-G_X(x))^{a-j}\,dx \tag{175}$$

$$= a^{\frac{1}{\beta}} \int_0^\delta \binom{a}{j} G_X(x)^j (1-G_X(x))^{a-j}\,dx + a^{\frac{1}{\beta}} \int_\delta^\infty \binom{a}{j} G_X(x)^j (1-G_X(x))^{a-j}\,dx. \tag{176}$$

We will analyze these two integral separately. It will turn out that the second integral vanishes as $a$ grows. □

*The first integral.* We have that

$$a^{\frac{1}{\beta}} \int_0^\delta \binom{a}{j} G_X(x)^j (1-G_X(x))^{a-j}\,dx \tag{177}$$

$$\le a^{\frac{1}{\beta}} \int_0^\delta \binom{a}{j} (1+\epsilon)^j \left(\frac{c}{\beta}\right)^j x^{\beta j} (1-(1-\epsilon)\frac{c}{\beta}x^\beta)^{a-j}\,dx \tag{178}$$

$$= \int_0^{\delta a^{\frac{1}{\beta}}} \binom{a}{j} (1+\epsilon)^j \left(\frac{c}{\beta}\right)^j \left(\frac{x}{a^{\frac{1}{\beta}}}\right)^{\beta j} \left(1-(1-\epsilon)\frac{c}{\beta}\left(\frac{x}{a^{\frac{1}{\beta}}}\right)^\beta\right)^{a-j}\,dx \tag{179}$$

$$= \int_0^{\delta a^{\frac{1}{\beta}}} \binom{a}{j} (1+\epsilon)^j \left(\frac{c}{\beta}\right)^j \frac{x^{\beta j}}{a^j} \left(1-(1-\epsilon)\frac{c}{\beta}\frac{x}{a}\right)^{a-j}\,dx. \tag{180}$$

Then

$$\int_0^{\delta a^{\frac{1}{\beta}}} \binom{a}{j} (1+\epsilon)^j \left(\frac{c}{\beta}\right)^j \frac{x^{\beta j}}{a^j} \left(1-(1-\epsilon)\frac{c}{\beta}\frac{x}{a}\right)^{a-j}\,dx = \int_0^\infty \phi_a(x)\,dx, \tag{181}$$

where

$$\phi_a(x) := \begin{cases} \binom{a}{j}(1+\epsilon)^j \left(\frac{c}{\beta}\right)^j \frac{x^{\beta j}}{a^j} \left(1-(1-\epsilon)\frac{c}{\beta}\frac{x}{a}\right)^{a-j}\,dx & \text{for } 0 \le x \le \delta a^{\frac{1}{\beta}} \\ 0 & \text{for } x > \delta a^{\frac{1}{\beta}}. \end{cases} \tag{182}$$

We have that

$$\lim_{a \to \infty} \phi_a(x) = \frac{1}{j!}(1+\epsilon)^j \left(\frac{c}{\beta}\right)^j x^{\beta j} e^{-(1-\epsilon)\frac{c}{\beta}x^\beta} \tag{183}$$

and

$$\phi_a(x) \le \frac{1}{j!}(1+\epsilon)^j \left(\frac{c}{\beta}\right)^j x^{\beta j} e^{-(1-\epsilon)\frac{c}{\beta}x^\beta} (1-(1-\epsilon)\frac{c}{\beta}\epsilon^\beta))^{-j} = C(j,\epsilon)x^{\beta j} e^{-(1-\epsilon)\frac{c}{\beta}x^\beta} \tag{184}$$

for a constant $C(j,\epsilon)$ independent of $a$. Now note that $\int_0^\infty x^{\beta j} e^{-(1-\epsilon)\frac{c}{\beta}x^\beta} < \infty$. It follows from the dominated convergence theorem that

$$\lim_{a \to \infty} \int_0^\infty \phi_a(x)\,dx = \int_0^\infty \lim_{a \to \infty} \phi_a(x)\,dx = \int_0^\infty \frac{1}{j!}(1+\epsilon)^j \left(\frac{c}{\beta}\right)^j x^{\beta j} e^{-(1-\epsilon)\frac{c}{\beta}x^\beta}\,dx < \infty. \tag{185}$$

Therefore, for $a$ sufficiently large,

$$\int_0^\delta \binom{a}{j} G_X(x)^j (1-G_X(x))^{a-j}\,dx \le a^{-\frac{1}{\beta}}(1+\epsilon)\int_0^\infty \frac{1}{j!}(1+\epsilon)^j \left(\frac{c}{\beta}\right)^j x^{\beta j} e^{-(1-\epsilon)\frac{c}{\beta}x^\beta}\,dx. \tag{186}$$

Analogously, we can show that for $a$ sufficiently large,

$$\int_0^\delta \binom{a}{j} G_X(x)^j (1 - G_X(x))^{a-j} \, dx \geq a^{-\frac{1}{\beta}} (1 - \epsilon) \int_0^\infty \frac{1}{j!} (1 - \epsilon)^j \left(\frac{c}{\beta}\right)^j x^{\beta j} e^{-(1+\epsilon)\frac{c}{\beta} x^\beta} \, dx. \tag{187}$$

Now observe that

$$\lim_{\epsilon \to 0^+} (1 + \epsilon) \int_0^\infty \frac{1}{j!} (1 + \epsilon)^j \left(\frac{c}{\beta}\right)^j x^{\beta j} e^{-(1-\epsilon)\frac{c}{\beta} x^\beta} \, dx \tag{188}$$

$$= \int_0^\infty \frac{1}{j!} \left(\frac{c}{\beta}\right)^j x^{\beta j} e^{-\frac{c}{\beta} x^\beta} \, dx \tag{189}$$

$$= \lim_{\epsilon \to 0^+} (1 - \epsilon) \int_0^\infty \frac{1}{j!} (1 - \epsilon)^j \left(\frac{c}{\beta}\right)^j x^{\beta j} e^{-(1+\epsilon)\frac{c}{\beta} x^\beta} \, dx, \tag{190}$$

where we once again apply the dominated convergence theorem. It follows that

$$\lim_{a \to \infty} \frac{\int_0^\delta \binom{a}{j} G_X(x)^j (1 - G_X(x))^{a-j} \, dx}{a^{-\frac{1}{\beta}}} = \int_0^\infty \frac{1}{j!} \left(\frac{c}{\beta}\right)^j x^{\beta j} e^{-\frac{c}{\beta} x^\beta} \, dx. \tag{191}$$

*The second integral.* We now analyze

$$a^{\frac{1}{\beta}} \int_\delta^\infty \binom{a}{j} G_X(x)^j (1 - G_X(x))^{a-j} \, dx. \tag{192}$$

Observe that

$$a^{\frac{1}{\beta}} \int_\delta^\infty \binom{a}{j} G_X(x)^j (1 - G_X(x))^{a-j} \, dx < a^{\frac{1}{\beta}} \binom{a}{j} \int_\delta^\infty (1 - G_X(x))^{a-j} \, dx \tag{193}$$

$$< a^{\frac{1}{\beta}} \binom{a}{j} (1 - G_X(\delta))^{a-j} \int_\delta^\infty 1 - G_X(x) \, dx \tag{194}$$

$$< a^{\frac{1}{\beta}} \binom{a}{j} (1 - G_X(\delta))^{a-j} \mathbb{E}[X]. \tag{195}$$

Thus,

$$\lim_{a \to \infty} \frac{\int_\delta^\infty \binom{a}{j} G_X(x)^j (1 - G_X(x))^{a-j} \, dx}{a^{\frac{1}{\beta}}} = 0. \tag{196}$$

Combining (191) and (196) gives us that

$$\int_0^\infty \binom{a}{j} G_X(x)^j (1 - G_X(x))^{a-j} \, dx \propto a^{-\frac{1}{\beta}}, \tag{197}$$

as desired.

*Proof of Theorem 4(iii).* Recall again that

$$h(a) := \sum_{i=1}^{\min\{k,a\}} \mu_{\mathcal{D}}(a - i + 1, a). \tag{198}$$

We show that $h$ is strictly concave and

$$\lim_{a \to \infty} h(a) - B \log a - C = 0 \tag{199}$$

for constants $B, C > 0$. Both of these facts follow directly from the lemma below. Theorem 4(iii) then follows immediately by applying Lemma D.1(iii).

LEMMA E.2. *For $\mathcal{D}$ an exponential distribution with rate parameter $\lambda$, so that $f_X(x) = \lambda e^{-\lambda x}$ for $\lambda > 0$,*

$$\lim_{a \to \infty} \mu_{\mathcal{D}}(a - i, a) - \log a - B(j) = 0 \tag{200}$$

*for a constant $B(j) > 0$. Moreover, $\mu_{\mathcal{D}}(a - i, a)$ is strictly concave.*

PROOF. For an exponential distribution with rate parameter $\lambda$, it is well known that

$$\mu_{\mathcal{D}}(a - i, a) = \sum_{j=i+1}^a \frac{1}{\lambda n}. \tag{201}$$

It is clear, then, that $\mu_{\mathcal{D}}(a - i, a)$ is strictly concave. (201) is equal to

$$\frac{1}{\lambda}\left(\log n + \gamma + \epsilon(a) - \sum_{j=1}^{i} \frac{1}{j}\right), \tag{202}$$

where $\gamma$ is the Euler-Mascheroni constant and $\lim_{a \to \infty} \epsilon(a) = 0$, from which (200) follows. □

*Proof of Theorem 4(iv).* Recall again that

$$h(a) := \sum_{i=1}^{\min\{k,a\}} \mu_{\mathcal{D}}(a - i + 1, a). \tag{203}$$

Then it suffices to show that $h$ is strictly concave and

$$\lim_{a \to \infty} \frac{h(a)}{Ba^{\frac{1}{\alpha}}} = 1 \tag{204}$$

for a constant $B > 0$. Both of these facts follow directly from the lemma below. Theorem 4(iv) then follows immediately by applying Lemma D.1(iv).

LEMMA E.3. *For $\mathcal{D}$ a Pareto distribution with pdf $f_X(x) = x^{-\alpha-1}$ for $\alpha > 1$,*

$$\lim_{a_t \to \infty} \frac{\mu_{\mathcal{D}}(a - i, a)}{a^{\frac{1}{\alpha}}} = C \tag{205}$$

*for a constant $C > 0$. Moreover, $\mu_{\mathcal{D}}(a - i, a)$ is strictly concave.*

PROOF. The result follows directly from Lemmas D.10 and D.11 in [29], where it is shown (in our notation) that

$$\lim_{a \to \infty} \frac{\mu_{\mathcal{D}}(a, a)}{a^{\frac{1}{\alpha}}} = \Gamma\left(\frac{\alpha - 1}{\alpha}\right) \tag{206}$$

and

$$\mu_{\mathcal{D}}(a - i, a) = \prod_{j=1}^{i}\left(1 - \frac{1}{j\alpha}\right)\mu_{\mathcal{D}}(a, a). \tag{207}$$

□

Thus,

$$\lim_{a \to \infty} \frac{\sum_{i=1}^{k} \mu_{\mathcal{D}}(a - i + 1, a)}{B \log a} = 1 \tag{208}$$

for a constant $B$. Also, note that $\mu_{\mathcal{D}}(a - i, a)$ is a constant multiple of $\mu_{\mathcal{D}}(a, a)$, and that $\mu_{\mathcal{D}}(a, a)$ is strictly concave, since the mean of the largest order statistic of a distribution is strictly concave in sample size. Thus, $\mu_{\mathcal{D}}(a - i, a)$ is strictly concave.

Received 20 February 2007; revised 12 March 2009; accepted 5 June 2009

