# OpenReview forum: "Reconciling the accuracy-diversity trade-off in recommendations"
_ACM.org/TheWebConf/2024/Conference — TheWebConf24 Oral_

### Official Review · Reviewer_y8ju · 2023-11-17

**Novelty:** 5
**Technical Quality:** 7

**Review:**

Summary
The paper studies the accuracy-diversity tradeoff in recommender systems. The paper claims that none of these widely studied metrics actaully capture the "utility" of recommendations to the end-usee and hence propose a new metric of user utility. They argue that accuracy is misaligned with user utility because it does not account for consumption constraints, and that utility-maximizing recommendations are naturally diverse due to diminishing returns of recommending similar items.
To provide a theoreical foundation for the user utility concept, the authors introduce a model where items belong to discrete types, and users have a probability distribution over types. In each session, the user is in the mood for one type, drawn from the distribution. The authors define a measure of diversity based on the representation of item types in a recommendation set, and derive asymptotic results for accuracy- and utility-maximizing recommendations as a function of model parameters.

Strengths
+ Overall the paper is extremely well written and easy to follow (even though the theoretical nature of the paper, this industry reviewer could understand it, barring a few details). The terminologies and formulations introduced are intuitive. The model structure presented is also easy to understand. The design choices are well motivated.
+ The accuracy-diversity tradeoff is a widely studied and consequential topic in recommender systems. A deeper and principled understanding of the tradeoff should be helpful to practitioners working in the filed.

Weakenesses
- The paper makes some strong claims without substantial backing. For instance, the claim that “Recommender systems are often built to maximize accuracy, the percentage of recommended items that a user likes.” is not entirely accurate. In practice, a variety of metrics like precision at a termination factor k are considered. These metrics take into account that the user can interact with only a limited number of item predictions. Furthermore, some works have incorporated the prediction of rare items which might pleasantly surprise the user. Propensity weighted precision[1] tried to ensure diversity by recommening rare items. A discussion about such metrics will help situate the paepr better.

[1]Extreme Multi-label Loss Functions for Recommendation, Tagging, Ranking & Other Missing Label Applications. Himanshu Jain, Yashoteja Prabhu, Manik Varma. Proceedings of the 22nd ACM SIGKDD International Conference on Knowledge Discovery and Data Mining (KDD '16), 935–944 (2016).

**Questions:**

Please refer to the weaknesses pointed out in the Review

**Reviewer Confidence:**

3: The reviewer is confident but not certain that the evaluation is correct

**Scope:**

4: The work is relevant to the Web and to the track, and is of broad interest to the community

---

### Official Review · Reviewer_WvZu · 2023-11-19

**Novelty:** 4
**Technical Quality:** 3

**Review:**

The paper explores the accuracy-diversity trade-off and defines some theorems to prove that accuracy is misaligned with user utility and diversity is aligned with user utility.

Strength:
The paper defines detailed theorems to systematically highlight that diversity should be incorporated to help align accuracy.

Weaknesses:

The paper explores the accuracy-diversity trade-off in a theorem manner to highlight the importance of the accuracy-diversity trade-off. However, it is well known that accuracy-diversity trade-off is important in recommender systems. I would be more interested in applying the theorem in a real recommendation problem, e.g., incorporating the trade-off to a recommendation algorithm to validate its recommendation performance.

The paper is not well explained. For example, Table 1 is not mentioned in the text. Theorem 1, Theorem 2a, Theorem 2b etc. appear before their definitions, which is hard to understand. The definition of user utility is also missing when user utility appears first time.

Related work is not included in the first 8 pages. At least it should be summarized and analyzed in the introduction.

I would suggest the authors summarize their primary contributions explicitly in the introduction.

The paper must state how the work is relevant to the Web and to the track on the first page.

**Questions:**

Weaknesses:

The paper explores the accuracy-diversity trade-off in a theorem manner to highlight the importance of the accuracy-diversity trade-off. However, it is well known that accuracy-diversity trade-off is important in recommender systems. I would be more interested in applying the theorem in a real recommendation problem, e.g., incorporating the trade-off to a recommendation algorithm to validate its recommendation performance.

The paper is not well explained. For example, Table 1 is not mentioned in the text. Theorem 1, Theorem 2a, Theorem 2b etc. appear before their definitions, which is hard to understand. The definition of user utility is also missing when user utility appears first time.

Related work is not included in the first 8 pages. At least it should be summarized and analyzed in the introduction.

I would suggest the authors summarize their primary contributions explicitly in the introduction.

The paper must state how the work is relevant to the Web and to the track on the first page.

**Reviewer Confidence:**

2: The reviewer is willing to defend the evaluation, but it is likely that the reviewer did not understand parts of the paper

**Scope:**

2: The connection to the Web is incidental, e.g., use of Web data or API

---

### Official Review · Reviewer_vP4g · 2023-11-22

**Novelty:** 6
**Technical Quality:** 5

**Review:**

The study explores the trade-off between accuracy and diversity in recommender systems when making recommendations. The authors introduce a third concept, user utility, arguing that accuracy alone is misaligned with user utility as it does not consider consumption constraints. The theoretical model demonstrates that recommendations maximizing user utility, by accounting for consumption constraints, naturally exhibit diversity, providing practical guidance for incorporating diversity to enhance user satisfaction in recommender systems.

Pros:
1.Overall, the draft is well-written and easy to follow.
2.It has its own novelty by revealing that accuracy might not be well-aligned metrics to user utility when we consider   users' consumption constraints. Furthermore, the author revealed that the proposed user utility and diversity are well aligned with each others.

Cons:
1.There are already some metrics to measure ranking quality that consider both  accuracy and diversity. One famous example is $\alpha$-ndcg, which I think should be discussed. But it is missed in this paper. I am wondering the connection and difference between proposed user utility and $\alpha$-ndcg. Can we have some mathematical interpretation of  $\alpha$-ndcg when we consider   users' consumption constraints. Will maximizing $\alpha$-ndcg trade-off diversity?
2. The introduction is too wordy which require too much of readers' patience. I suggest the authors to reorganize it.  For example, the authors can save some detailed conclusion to the Result section.
3.It would be better to put the notation table earlier, say in Section 2.
4. Codes is not shared for reproduction.

**Questions:**

1. As I mentioned,  I am wondering the connection and difference between proposed user utility and $\alpha$-ndcg. Can we have some mathematical interpretation of  $\alpha$-ndcg when we consider   users' consumption constraints. Will maximizing $\alpha$-ndcg trade-off diversity? It would be better to have some discussion of it in the paper.
2. In eq.32, what is $V_{t,i}, k$?

**Reviewer Confidence:**

3: The reviewer is confident but not certain that the evaluation is correct

**Scope:**

4: The work is relevant to the Web and to the track, and is of broad interest to the community

---

### Official Review · Reviewer_epgH · 2023-11-23

**Novelty:** 5
**Technical Quality:** 4

**Review:**

This paper explores the balance between accuracy and diversity in recommender systems, emphasizing the importance of user utility. It argues that accuracy, traditionally emphasized in these systems, often overlooks user consumption constraints, leading to recommendations that lack diversity. The study suggests that incorporating diversity, considering user constraints, not only enhances user utility but also aligns recommendations with users' diverse interests, challenging the conventional accuracy-focused approach in recommender systems​​​​.

Strong points:

1. This paper emphasizes that traditional accuracy metrics in recommender systems do not fully capture user utility. By considering user consumption constraints, such as the limited number of items a user can engage with at a time, the paper argues for a more user-centric approach that aligns recommendations with actual user needs and preferences.
2. The paper's theoretical model demonstrates that utility-maximizing recommendations naturally lead to diversity due to diminishing returns of similar items.

Weak points:
1. This paper does not address a unique problem, its novelty could be considered limited.
2. The paper could have limitations in its technical depth or innovation.
3. According to the formatting requirements of the web conference, the first 8 pages should be self-contained, since reviewers are not required to read past that. The amount of additional content seems excessive and could potentially overwhelm or distract from the main findings of the paper. I suggest considering a reduction or more focused selection of these materials to enhance the clarity and impact of the research presented.

**Questions:**

How do you quantify user utility in your model, and what factors are considered in its calculation? Can this model adapt to different types of users with varying preferences and consumption patterns?

**Ethics Review Description:**

NA.

**Reviewer Confidence:**

3: The reviewer is confident but not certain that the evaluation is correct

**Scope:**

4: The work is relevant to the Web and to the track, and is of broad interest to the community

---

### Official Review · Reviewer_XkB1 · 2023-11-24

**Novelty:** 6
**Technical Quality:** 5

**Review:**

This paper inspects the trade-off between fairness and accuracy in recommender system. It brings a new concept: user utility. And by incorporating consumption constraints into the theoretical model, the authors demonstrate that utility-maximizing recommendations naturally exhibit diversity. They show that as the number of recommendations increases, there are diminishing returns in recommending similar items, leading to a preference for diverse recommendations.
The authors provide theoretical proofs to support their arguments and empirically validate the accuracy of their theoretical claims.

The view is interesting, and the paper is well-structured and well-written. However, I have a question regarding the definition of utility and its impact on theoretical results. Currently, utility is defined as the probability of a user liking at least one item in the recommender set. In reality, user satisfaction increases with the number of liked items, and our goal is to maximize long-term user satisfaction. How will this affect the theoretical results?

Could the authors please provide additional insights into why setting alpha=1 leads to inferior outcomes as demonstrated in Table 2? Furthermore, how does this impact the validity of the derived results in real-world recommendation algorithms?

**Questions:**

Please refer to above section.

**Reviewer Confidence:**

3: The reviewer is confident but not certain that the evaluation is correct

**Scope:**

3: The work is somewhat relevant to the Web and to the track, and is of narrow interest to a sub-community

---

### Decision · Program_Chairs · 2024-01-22

**Decision:**

Accept (Oral)

**Comment:**

Summary: The paper looks into the trade-off between fairness and accuracy in recommender system.

 Strengths:
 + introduces new concept of user utility and motivates how utility-maximizing recommendation exhibits diversity
 + rigorous theoretical treatment
 + empirical validation of said claims

 Weaknesses:
 - lack of comparison to existing metrics on diversity
 - related work should be part of the main paper
 - some concern about length of appendix as final paper has limit

 Recommendation: Accept. Theoretical treatment of a problem of broad interest.